# Robust and Conjugate Spatio-Temporal Gaussian Processes

**William Laplante** [1 2 3] **Matias Altamirano** [2] **Andrew Duncan** [3 4] **Jeremias Knoblauch** [2] **François-Xavier Briol** [2]

## Abstract

State-space formulations allow for Gaussian process (GP) regression with linear-in-time computational cost in spatio-temporal settings, but performance typically suffers in the presence of outliers. In this paper, we adapt and specialise the *robust and conjugate GP (RCGP)* framework of Altamirano et al. (2024) to the spatio-temporal setting. In doing so, we obtain an outlier-robust spatio-temporal GP with a computational cost comparable to classical spatio-temporal GPs. We also overcome the three main drawbacks of RCGPs: their unreliable performance when the prior mean is chosen poorly, their lack of reliable uncertainty quantification, and the need to carefully select a hyperparameter by hand. We study our method extensively in finance and weather forecasting applications, demonstrating that it provides a reliable approach to spatio-temporal modelling in the presence of outliers.

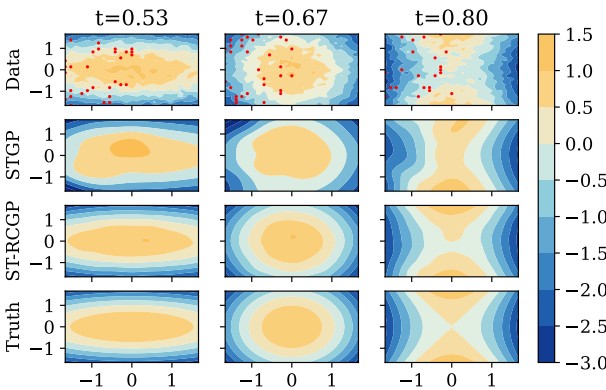

*Figure 1. Spatio-temporal GPs in the presence of outliers.* The top row shows the observed data as a function of spatial covariates $\mathbf{s} \in \mathbb{R}^2$. Outliers are highlighted in red and are uniformly distributed on $[-8, -6] \cup [6, 8]$. The second row gives the fit of a regular STGP, whilst the third row gives our proposed ST-RCGP fit. The last row shows the true latent function $f(s_1, s_2, t) = \sin(2\pi t)s_1^2 + \cos(2\pi t)s_2^2$. Further details are provided in Appendix C.4.

## 1. Introduction

Gaussian processes (GPs; Williams & Rasmussen (2006)) are flexible probabilistic models used in a vast class of problems from regression (Williams & Rasmussen, 2006) to emulation (Santner et al., 2018) and optimisation (Garnett, 2023). GPs originated in spatial statistics, where their use for regression was known as kriging (Krige, 1951; Stein, 1999), but they have more recently been used widely in spatio-temporal settings, including in epidemiology (Senanayake et al., 2016), neuroimaging (Hyun et al., 2016), object tracking (Aftab et al., 2019), and psychological studies (Kupilik

---
[1]Department of Physics and Astronomy, University College London, London, United Kingdom [2]Department of Statistical Science, University College London, London, United Kingdom [3]The Alan Turing Institute, London, United Kingdom [4]Department of Mathematics, Imperial College London, London, United Kingdom. Correspondence to: William Laplante <william.laplante.24@ucl.ac.uk>.

*Proceedings of the $42^{nd}$ International Conference on Machine Learning*, Vancouver, Canada. PMLR 267, 2025. Copyright 2025 by the author(s).

& Witmer, 2018). Their popularity arises from their ability to encode spatial and temporal properties such as smoothness, periodicity, and sparsity (Duvenaud, 2014), allowing them to model phenomena such as local weather patterns or seasonality. Crucially, GPs also have an exact, closed-form posterior when using a Gaussian likelihood. However, naive implementations have a cubic computational cost in the number of data points $N$, limiting their scalability. To address this issue, a plethora of approximations have been proposed (Drineas et al., 2005; Titsias, 2009; Hensman et al., 2013; Wilson & Nickisch, 2015). While effective, these do not typically recover the exact GP, require careful tuning, and can degrade performance for complex datasets (Bauer et al., 2016; Pleiss et al., 2018).

In spatio-temporal settings, an alternative strategy is to reformulate the GP as a state-space model (SSM) (Reece & Roberts, 2010; Hartikainen & Särkkä, 2010; Sarkka & Hartikainen, 2012; Solin, 2016; Nickisch et al., 2018; Hamelijnck et al., 2021). This gives rise to a class of models—*spatio-temporal Gaussian processes (STGPs)*—with linear cost in the number of temporal observations. But, as with standard GPs, STGPs lack robustness to model misspecification, such as with outliers arising from extreme events

(Heaton et al., 2011), measurement errors that are spatially correlated (Tadayon & Rasekh, 2019), and other heterogeneities (Fonseca et al., 2023). Figure 1 illustrates this in the case of outliers. Clearly, the STGP (2nd row) fails to align with the ground truth (4th row).

To resolve this issue, existing work on STGPs has focused on using likelihoods corresponding to distributions which are more expressive than Gaussians, such as mixtures or heavy-tailed distributions. Doing so breaks conjugacy, and the posterior must typically be approximated (Hartikainen et al., 2011; Solin & Särkkä, 2014; Hamelijnck et al., 2021). We refer the reader to Nickisch et al. (2018) for a comprehensive overview and to Wilkinson et al. (2023) for a Python package providing those algorithms. Although these methods have been efficiently implemented, they typically use additional optimisation steps at each time-point and are hence significantly more costly than conjugate STGPs. We also highlight a small body of work that uses outlier-rejection Kalman filters with STGPs to improve robustness in inference tasks (Bock et al., 2022; Waxman & Djurić, 2024). While these outlier-rejection methods offer robustness at lower computational cost compared to non-conjugate STGPs, they are generally considered less expressive and not as strongly supported by theoretical foundations.

Recently, Altamirano et al. (2024) introduced a method, called *robust and conjugate Gaussian processes (RCGPs)*, that uses generalised Bayesian inference (Bissiri et al., 2016; Knoblauch et al., 2022) to confer robustness to standard GPs. Their approach is highly attractive since it provably provides robustness to outliers whilst maintaining conjugacy, but it shares the cubic cost of GPs. Existing work on RCGPs also has three main limitations: RCGPs underperform when the prior mean is chosen poorly (see Ament et al. (2024) and the Appendix of Altamirano et al. (2024)), their uncertainty quantification properties have not been well-studied, and they have an additional hyperparameter compared to standard GPs. Further, the existing heuristic for tuning this parameter relates to the proportion of outliers in the data, which is unknown in practice and generally has to be hand-picked on a case-by-case basis.

In this paper, we show how to refine and specialise the RCGP framework for spatio-temporal data. Our algorithm, denoted *spatio-temporal RCGP (ST-RCGP)*, inherits the computational and memory efficiency of STGPs, as well as the robustness properties of RCGPs (see 3rd row of Figure 1). The sequential aspect of the state-space formulation also allows us to overcome the three main limitations of RCGPs (sensitivity to prior mean, lack of reliable uncertainty quantification, and the additional hyperparameter). Overall, we observe that the ST-RCGP provides inferences comparable to state-of-the-art non-Gaussian STGPs with a computational cost similar to classical STGPs.

## 2. Background

**GP Regression** Let $\{\mathbf{x}_k, y_k\}_{k=1}^N$ be observations, where $\mathbf{x}_k \in \mathcal{X} \subseteq \mathbb{R}^d$ are covariates and $y_k \in \mathcal{Y} \subseteq \mathbb{R}$ are responses. For observation noise $\epsilon_k$, GP regression considers

$$y_k = f(\mathbf{x}_k) + \epsilon_k, \quad \text{for } k = 1, \ldots, N \tag{1}$$

where the latent function $f : \mathcal{X} \to \mathbb{R}$ is modelled by a GP prior $f \sim \mathcal{GP}(m, \kappa)$ with mean $m : \mathcal{X} \to \mathbb{R}$ and covariance (or kernel) $\kappa : \mathcal{X} \times \mathcal{X} \to \mathbb{R}$, such that for any inputs $\mathbf{X} = (\mathbf{x}_1, ..., \mathbf{x}_N)^\top$, the vector $\mathbf{f} = (f(\mathbf{x}_1), ..., f(\mathbf{x}_N))^\top$ is distributed as a $N$-dimensional Gaussian $\mathcal{N}(\mathbf{m}, \mathbf{K})$ with $\mathbf{m} = (m(\mathbf{x}_1), ..., m(\mathbf{x}_N))^\top$ and $(\mathbf{K})_{ij} = \kappa(\mathbf{x}_i, \mathbf{x}_j)$. The kernel $\kappa$ typically depends on parameters $\boldsymbol{\theta} \in \Theta$ that we omit from the notation for brevity. When the observations $\mathbf{y} = (y_1, ..., y_N)^\top$ have independent Gaussian noise $(\epsilon_1, \ldots, \epsilon_N)^\top \sim \mathcal{N}(0, \sigma^2 \mathbf{I}_N)$, the posterior predictive for $f_\star = f(x_\star)$ at $x_\star \in \mathcal{X}$ is $f_\star | \mathbf{y}, \mathbf{X} \sim \mathcal{N}(\mu_{\text{GP}}^\star, \Sigma_{\text{GP}}^\star)$ with

$$\begin{aligned} \mu_{\text{GP}}^\star &= m_\star + \mathbf{k}_\star^\top \left(\mathbf{K} + \sigma^2 \mathbf{I}_N\right)^{-1} (\mathbf{y} - \mathbf{m}) \\ \Sigma_{\text{GP}}^\star &= k_{\star\star} - \mathbf{k}_\star^\top (\mathbf{K} + \sigma^2 \mathbf{I}_N)^{-1} \mathbf{k}_\star \end{aligned} \tag{2}$$

where $\mathbf{I}_N$ is the $N \times N$ identity matrix, $\mathbf{k}_\star = (\kappa(x_\star, x_1), ..., \kappa(x_\star, x_N))^\top$, $k_{\star\star} = \kappa(x_\star, x_\star)$, and $m_\star = m(x_\star)$. The orange colouring can be ignored for now, but will be used to highlight differences with RCGPs. To obtain this mean and covariance, we must invert an $N \times N$ matrix—an operation with computational complexity $\mathcal{O}(N^3)$.

**State-Space Formulation** An alternative approach, which has linear-in-time cost, is to use a state-space representation. Consider Equation (1) with spatio-temporal inputs. At time $t_k \in \mathcal{T} \subseteq \mathbb{R}$, we now have inputs $\mathbf{x}_{k,j} = (\mathbf{s}_j, t_k) \in \mathcal{X} = \mathcal{S} \times \mathcal{T}$ on a spatial grid $\mathbf{S} = (\mathbf{s}_1, \ldots, \mathbf{s}_{n_s})^\top \in \mathcal{S}^{n_s} \subseteq \mathbb{R}^{n_s \times d_s}$ with $n_s$ points and spatial dimensionality $d_s$ (i.e. $d = d_s + 1$). The observations corresponding to $\mathbf{x}_k = (\mathbf{x}_{k,1}, \ldots, \mathbf{x}_{k,n_s})^\top$ are denoted $\mathbf{y}_k = (y_{k,1}, \ldots, y_{k,n_s})^\top \in \mathbb{R}^{n_s}$, leading to a total number of data points $N = n_t n_s$ where $n_t$ is the number of time steps. If standard GPs were used here, the cost would be $\mathcal{O}(n_t^3 n_s^3)$, which may become impractical.

A solution to this issue is to reformulate GPs as SSMs. We first collect $\nu$ partial derivatives of $f$ with respect to time in a state vector $\mathbf{z} = \mathbf{z}(\mathbf{s}, t) \in \mathbb{R}^{(\nu+1)}$ where $(\mathbf{z}(\mathbf{s}, t))_i = \frac{\partial^{i-1}}{\partial t^{i-1}} f(\mathbf{s}, t)$ for $i = 1, ..., \nu+1$. We assume a stationary and separable kernel so that $\kappa(\mathbf{s}, t; \mathbf{s}', t') = \kappa_s(\mathbf{s} - \mathbf{s}') \kappa_t(t - t')$, where $\kappa_s$ and $\kappa_t$ are spatial and temporal kernels respectively. For a large class of kernels (Solin, 2016), we can represent the GP prior as the solution to a stochastic differential equation (SDE):

$$\frac{\partial \mathbf{z}(\mathbf{s}, t)}{\partial t} = \mathbf{F}_t \mathbf{z}(\mathbf{s}, t) + \mathbf{L}_t \mathbf{w}(\mathbf{s}, t), \tag{3}$$

where $\mathbf{F}_t \in \mathbb{R}^{(\nu+1)\times(\nu+1)}$, $\mathbf{L}_t \in \mathbb{R}^{(\nu+1)\times 1}$, and $\mathbf{w}(\mathbf{s},t)$ is a spatio-temporal white noise process corresponding to the derivative of Brownian motion with spectral density $\mathbf{Q}_{c,t} \in \mathbb{R}^{1\times(\nu+1)}$ (Solin, 2016). The matrices $\mathbf{F}_t$, $\mathbf{L}_t$, $\mathbf{Q}_{c,t}$ and the constant $\nu$ depend on $\kappa$. Equation (3) defines a continuous latent process, but given a finite collection of points, this becomes a SSM with states $\mathbf{z}_k = (\mathbf{z}(\mathbf{s}_1,t_k),...,\mathbf{z}(\mathbf{s}_{n_s},t_k))^\top \in \mathbb{R}^{n_s(\nu+1)}$ and time steps $\Delta t_k = t_k - t_{k-1}$ (Hartikainen & Särkkä, 2010; Sarkka & Hartikainen, 2012; Särkkä & Solin, 2019):

$$\mathbf{z}_0 \sim \mathcal{N}(\mathbf{0}, \boldsymbol{\Sigma}_0), \text{ and for } k > 0,$$
$$\mathbf{z}_k = \mathbf{A}_{k-1}\mathbf{z}_{k-1} + \mathbf{q}_{k-1}, \quad \mathbf{q}_{k-1} \sim \mathcal{N}(\mathbf{0}, \boldsymbol{\Sigma}_{k-1}) \quad (4)$$
$$\mathbf{y}_k = \mathbf{H}\mathbf{z}_k + \epsilon_k$$

where $\mathbf{H} \in \mathbb{R}^{n_s \times n_s(\nu+1)}$ is defined such that $\mathbf{H}\mathbf{z}_k = \mathbf{f}_k := (f(\mathbf{s}_1,t_k),...,f(\mathbf{s}_{n_s},t_k))^\top$. The matrices $\mathbf{A}_{k-1}$ and $\boldsymbol{\Sigma}_{k-1}$ depend on $\mathbf{F}_t$, $\mathbf{L}_t$ and $\mathbf{Q}_{c,t}$ in the SDE formulation. We define them in Appendix C.2, and provide information on how to compute them in practice for a list of common kernels $\kappa$.

**Filtering and Smoothing**   Solving the SSM from Equation (4) via sequential inference amounts to first retrieving the filtering distribution $p(\mathbf{z}_k|\mathbf{y}_{1:k})$, and then the smoothing distribution $p(\mathbf{z}_k|\mathbf{y}_{1:N})$ (Särkkä, 2013). Taking Equation (4), and assuming the observation noise is Gaussian so that $\epsilon_k \sim \mathcal{N}(0,\sigma^2\mathbf{I}_{n_s})$, the predict and update equations are *conjugate* and given by the Kalman filter (Kalman, 1960) and Rauch-Tung-Striebel smoother (Rauch et al., 1965); see Section 8.2 of Särkkä (2013). The resulting distribution has densities $p(\mathbf{z}_k|\mathbf{y}_{1:k-1}) = \mathcal{N}(\mathbf{z}_k; \mathbf{m}_{k|k-1}, \mathbf{P}_{k|k-1})$ and $p(\mathbf{z}_k|\mathbf{y}_{1:k}) = \mathcal{N}(\mathbf{z}_k; \mathbf{m}_{k|k}, \mathbf{P}_{k|k})$, with *predict step*:

$$\mathbf{m}_{k|k-1} := \mathbf{A}_{k-1}\mathbf{m}_{k-1|k-1}$$
$$\mathbf{P}_{k|k-1} := \mathbf{A}_{k-1}\mathbf{P}_{k-1|k-1}\mathbf{A}_{k-1}^\top + \boldsymbol{\Sigma}_{k-1}, \quad (5)$$

and *(Bayes) update step*:

$$\mathbf{P}_{k|k} := \left(\mathbf{P}_{k|k-1}^{-1} + \mathbf{H}^\top \sigma^{-2}\mathbf{I}_{n_s}\mathbf{H}\right)^{-1}$$
$$\mathbf{K}_k := \mathbf{P}_{k|k}\mathbf{H}^\top \sigma^{-2}\mathbf{I}_{n_s} \quad (6)$$
$$\mathbf{m}_{k|k} := \mathbf{m}_{k|k-1} + \mathbf{K}_k(\mathbf{y}_k - \hat{\mathbf{f}}_k),$$

where $\hat{\mathbf{f}}_k := \mathbf{H}\mathbf{m}_{k|k-1}$, $\mathbf{K}_k \in \mathbb{R}^{n_s(\nu+1)\times n_s}$, $\mathbf{m}_{k|k} \in \mathbb{R}^{n_s(\nu+1)}$ and $\mathbf{P}_{k|k} \in \mathbb{R}^{n_s(\nu+1)\times n_s(\nu+1)}$. The posterior $p(\mathbf{z}_k|\mathbf{y}_{1:N})$ is obtained by marginalising the smoothing distribution, and matches exactly the original GP posterior (Solin, 2016). For a new input $(\mathbf{s}_\star, t_\star)$, the predictive is obtained by including $(\mathbf{s}_\star, t_\star)$ in the filtering and smoothing algorithms and predicting without updating. Importantly, this algorithm only inverts matrices whose sizes scale with $\nu$—where typically $\nu < 10$ (Hartikainen & Särkkä, 2010)—or $n_s$, but *not* with $n_t$. In contrast to the standard GP's $\mathcal{O}(n_t^3 n_s^3)$ time and $\mathcal{O}(n_t^2 n_s^2)$ memory cost, the STGP only requires $\mathcal{O}(n_t n_s^3)$ time and $\mathcal{O}(n_t n_s^2)$ memory (Solin, 2016).

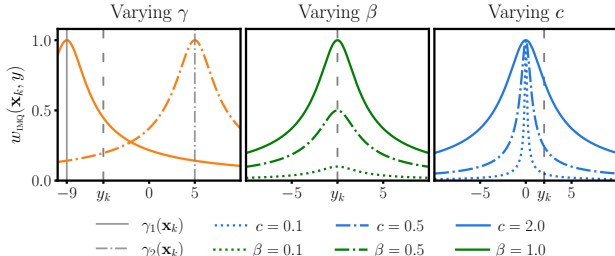

*Figure 2. Behaviour of $y \mapsto w_{\mathrm{IMQ}}(\mathbf{x}_k, y)$ as we vary $c$, $\beta$, and $\gamma$. We emphasize $|y_k - \gamma(\mathbf{x}_k)|$ and $w_{\mathrm{IMQ}}(\mathbf{x}_k, y_k)$ with datum $y_k$.*

**Generalised Bayes for GP Regression**   Conjugate GPs and STGPs rely on the fragile assumption that $\epsilon_1,...,\epsilon_N$ are Gaussian. While non-Gaussian likelihoods can address this, they also break conjugacy—and thus require expensive and potentially inaccurate approximations. Fortunately, generalised Bayes (GB) (Bissiri et al., 2016; Knoblauch et al., 2022) can produce robust and conjugate GP posteriors. GB mitigates model misspecification by constructing posterior distributions through a loss $\mathcal{L} : \mathcal{Y}^N \times \mathcal{Y}^N \times \mathcal{X}^N \to \mathbb{R}$ via

$$p_{\mathcal{L}}(\mathbf{f}|\mathbf{X}, \mathbf{y}) \propto p(\mathbf{f}|\mathbf{X})\exp\left(-N\mathcal{L}(\mathbf{y}; \mathbf{f}, \mathbf{X})\right), \quad (7)$$

where $\propto$ denotes proportionality and $\mathcal{L}$ is typically an estimator of a statistical divergence between the likelihood model and the data (Jewson et al., 2018). The loss $\mathcal{L}$ used for RCGPs (Altamirano et al., 2024) is a modification of the *weighted score-matching divergence*, first proposed by Barp et al. (2019), to regression. It generalises score-matching (Hyvärinen, 2005), and is a special case of the framework in Lyu (2009); Yu et al. (2019); Xu et al. (2022). Importantly, it uses a weight function $w : \mathcal{X} \times \mathcal{Y} \to \mathbb{R} \setminus \{0\}$ that down-weights unreasonably extreme data points. This choice yields the RCGP posterior predictive $f_\star \sim \mathcal{N}(\mu_{\mathrm{RCGP}}^\star, \Sigma_{\mathrm{RCGP}}^\star)$ with

$$\mu_{\mathrm{RCGP}}^\star = m_\star + \mathbf{k}_\star^\top\left(\mathbf{K} + \sigma^2\mathbf{J_w}\right)^{-1}(\mathbf{y} - \mathbf{m_w})$$
$$\Sigma_{\mathrm{RCGP}}^\star = k_{\star\star} - \mathbf{k}_\star^\top(\mathbf{K} + \sigma^2\mathbf{J_w})^{-1}\mathbf{k}_\star, \quad (8)$$

where $\mathbf{w} = (w(\mathbf{x}_1, y_1),...,w(\mathbf{x}_N, y_N))^\top$, and

$$\mathbf{m_w} = \mathbf{m} + \sigma^2\nabla_y\log(\mathbf{w}^2)$$
$$\mathbf{J_w} = \mathrm{diag}\left(\frac{\sigma^2}{2}\mathbf{w}^{-2}\right).$$

The expressions look similar to those in Equation (2), with differences highlighted in orange. This also reveals that standard GPs are recovered for the constant weight $w(\mathbf{x}, y) = \sigma/\sqrt{2}$. The similar forms also suggest that, much like classical GPs, a naive implementation of RCGPs has a complexity of $\mathcal{O}(N^3)$ due to the associated matrix inversion. However, under mild conditions on $w$, RCGPs benefit from provable robustness to outliers (see Proposition 3.2 of Altamirano et al., 2024).

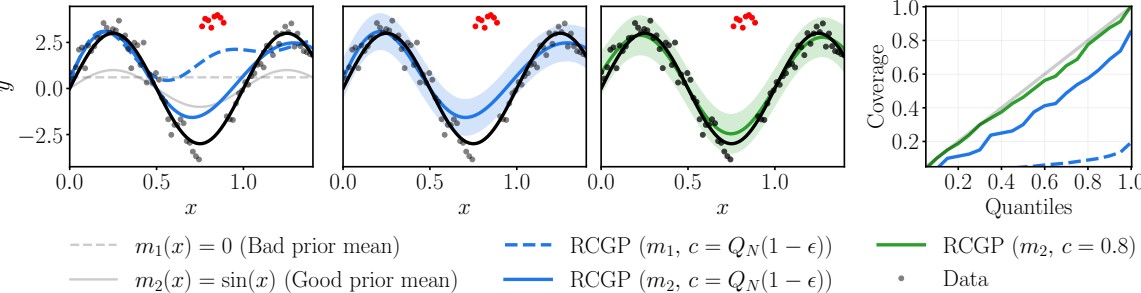

Figure 3. *Failure Modes with Existings RCGPs.* We generated $N = 80$ data points from the true function $f(x) = 3\sin(2\pi x)$ using $\mathcal{N}(0, 0.5)$ additive noise, then corrupted $\epsilon = 10\%$ of observations around $x = 0.76$. The leftmost figure shows that RCGP's posterior mean is strongly affected by the choice of prior mean, highlighting **Issue #1**. In the two middle plots, we demonstrate that the recommended $c = Q_N(1 - \epsilon) = 3.7$ (from Altamirano et al. (2024)) is *far* from optimal—emphasizing **Issue #3**—even when fitting simple data and knowing the true proportion of outliers $\epsilon$. These two plots also show RCGP's inaccurate uncertainty quantification when hyperparameters are chosen poorly, which is further supported by the rightmost coverage plot and illustrates **Issue #2** .

In prior work, Altamirano et al. (2024) suggested to take $w$ of the form of an inverse multi-quadric (IMQ) kernel depending on a *centering function* $\gamma : \mathcal{X} \to \mathbb{R}$, a *shrinking function* $c : \mathcal{X} \to \mathbb{R}$, and a constant $\beta > 0$:

$$w_{\text{IMQ}}(\mathbf{x}, y) = \beta \left( 1 + \frac{(y - \gamma(\mathbf{x}))^2}{c(\mathbf{x})^2} \right)^{-\frac{1}{2}} \quad (9)$$

The IMQ has been used frequently to robustify score-based divergences (Barp et al., 2019; Key et al., 2021; Matsubara et al., 2022; Altamirano et al., 2023; Matsubara et al., 2024; Liu & Briol, 2024). It is a bump function centred at $\gamma(\mathbf{x})$ that grows as $|y - \gamma(\mathbf{x})|$ decreases, and shrinks as $|y - \gamma(\mathbf{x})|$ increases (see Figure 2). The centering function $\gamma(\mathbf{x})$ determines where the bump is maximised, and any $y$ far from $\gamma(\mathbf{x})$ is down-weighted (see Figure 2). The shrinking function $c(\mathbf{x})$ determines how quickly we down-weight observations that deviate from $\gamma(\mathbf{x})$, and $\beta$ determines the maximum of the weight function (see Figure 2). Note that $\beta$ is a multiplicative constant which is equivalent to a GB learning rate (Wu & Martin, 2023).

While IMQ-based weights have shown much promise, they rely on three hyperparameters that can be difficult to select: $\gamma$, $\beta$, and $c$. Altamirano et al. (2024) proposed centering at the prior mean via $\gamma(\mathbf{x}) = m(\mathbf{x})$, and setting $\beta = \sigma/\sqrt{2}$ to guarantee that observations are never assigned a larger weight than they would in a standard GP. Finally, they suggested a heuristic choice for $c$ based on the assumed proportion of outliers: if $\epsilon \in [0, 1]$ is the expected proportion of outliers, they suggest setting the shrinking function to the constant $c(\mathbf{x}) = Q_N(1 - \epsilon)$, for $Q_N(1 - \epsilon)$ the $(1 - \epsilon)^{\text{th}}$ quantile of $\{|y_k - \gamma(\mathbf{x}_k)|\}_{k=1}^{N}$.

These choices can provide good performance in many settings, but introduce several failure modes:

**Issue #1 (Sensitivity to the prior mean $m$):** Altamirano et al. (2024); Ament et al. (2024) pointed to the poor perfor-

mance of RCGPs when $m$ is not chosen carefully. Indeed, setting $\gamma(\mathbf{x}) := m(\mathbf{x})$ can be undesirable when $m$ is not a good approximation of $f$. In that case, observations close to $m(\mathbf{x})$ but far from $f(\mathbf{x})$ will have large weights despite likely being outliers, whereas observations close to $f(\mathbf{x})$ but far from $m(\mathbf{x})$ will be down-weighted despite likely having little noise. We show this in Figure 3. Altamirano et al. (2024) suggest fixing this problem by choosing a prior mean using a simpler regression model, but this solution requires an additional fit of the data, which can be cumbersome.

**Issue #2 (Poor uncertainty quantification):** The values of $\beta$ and $c$ have a significant influence on the predictive variance, but it is not clear that the suggested choices are sensible when it comes to uncertainty quantification, and Altamirano et al. (2024) did not study this question. This could mean that RCGPs are consistently under- or overconfident—a problem we also illustrate in Figure 3. Sinaga et al. (2024) proposed a computation-aware extension of RCGPs to improve uncertainty quantification, but their approach still depends on having good values for $\beta$ and $c$.

**Issue #3 (Selection of shrinking function $c$):** The proposed heuristic for selecting $c$ requires the user to speculate on the proportion $\epsilon$ of outliers. Not only is this difficult to do reliably if we do not know $\epsilon$, but, as shown in Figure 3, the approach can also fail to select good values of $c$ even when the correct $\epsilon$ is used. This can lead to unreliable posterior estimates and either under- or overconfidence in the RCGP posterior. In addition, setting $\mathbf{x} \mapsto c(\mathbf{x})$ to be constant can be sub-optimal when outliers are clustered in time or space, in which case adapting the shrinking function according to its input could lead to improved uncertainty quantification.

## 3. Methodology

**Spatio-Temporal RCGPs** We now show that, similarly to GPs, inference for RCGPs can be performed using an SSM

representation and filtering/smoothing updates. To achieve this, we use the weighted score-matching loss

$$\mathcal{L}(\mathbf{x}_k, \mathbf{y}_k, \mathbf{z}_k) = \frac{1}{n_s} \sum_{j=1}^{n_s} \|w(\mathbf{x}_{k,j}, y_{k,j}) s_{f_{k,j}}(y_{k,j})\|_2^2$$
$$+ 2\nabla_y \cdot \left( w(\mathbf{x}_{k,j}, y_{k,j})^2 s_{f_{k,j}}(y_{k,j}) \right), \quad (10)$$

where $s_f(y) := (f - y)/\sigma$ is the score of the Gaussian error, and the dependency on $\mathbf{z}_k$ is through $f_{k,j} := (\mathbf{H}\mathbf{z}_k)_j$. Then, the generalised filtering posterior equations are

$$p_{\mathcal{L}}(\mathbf{z}_k|\mathbf{y}_{1:k}) \propto p_{\mathcal{L}}(\mathbf{z}_k|\mathbf{y}_{1:k-1}) \exp\left(-n_s \mathcal{L}(\mathbf{x}_k, \mathbf{y}_k, \mathbf{z}_k)\right),$$
$$p_{\mathcal{L}}(\mathbf{z}_k|\mathbf{y}_{1:k-1}) = \int p(\mathbf{z}_k|\mathbf{z}_{k-1}) p_{\mathcal{L}}(\mathbf{z}_{k-1}|\mathbf{y}_{1:k-1}) d\mathbf{z}_{k-1},$$

for $p(\mathbf{z}_k|\mathbf{z}_{k-1})$ defined as for Equation (4).

**Proposition 3.1.** *Suppose we model $\epsilon_1, \ldots, \epsilon_N \overset{iid}{\sim} \mathcal{N}(0, \sigma^2)$ and choose $w : \mathcal{X} \times \mathcal{Y} \to \mathbb{R} \setminus \{0\}$ such that $y \mapsto w(y, x)$ is differentiable. Then, the generalised posterior $p_{\mathcal{L}}(\mathbf{z}_k|\mathbf{y}_{1:k})$ associated to the SSM formulation of RCGPs is a $\mathcal{N}(\mathbf{z}_k; \mathbf{m}_{k|k}^{GB}, \mathbf{P}_{k|k}^{GB})$ and inference can be performed through Kalman filtering/smoothing with predict step*

$$\mathbf{m}_{k|k-1} := \mathbf{A}_{k-1} \mathbf{m}_{k-1|k-1}^{GB},$$
$$\mathbf{P}_{k|k-1} := \mathbf{A}_{k-1} \mathbf{P}_{k-1|k-1}^{GB} \mathbf{A}_{k-1}^{\top} + \mathbf{\Sigma}_{k-1},$$

*and update step*

$$\mathbf{P}_{k|k}^{GB} := \left( \mathbf{P}_{k|k-1}^{-1} + \mathbf{H}^{\top} \sigma^{-2} \mathbf{J}_{\mathbf{w}_k}^{-1} \mathbf{H} \right)^{-1},$$
$$\mathbf{K}_k^{GB} := \mathbf{P}_{k|k}^{GB} \mathbf{H}^{\top} \sigma^{-2} \mathbf{J}_{\mathbf{w}_k}^{-1},$$
$$\mathbf{m}_{k|k}^{GB} := \mathbf{m}_{k|k-1} + \mathbf{K}_k^{GB}(\mathbf{y}_k - \hat{\mathbf{f}}_{w_k}),$$

*where $\hat{\mathbf{f}}_{w_k} := \hat{\mathbf{f}}_k + \sigma^2 \nabla_y \log(\mathbf{w}_k^2)$ and $\mathbf{J}_{\mathbf{w}_k} := diag(\frac{\sigma^2}{2}\mathbf{w}_k^{-2})$. Moreover, the associated predictive is*

$$p_{\mathcal{L}}(\mathbf{y}_k|\mathbf{y}_{1:k-1}) = \int p(\mathbf{y}_k|\mathbf{z}_k) p_{\mathcal{L}}(\mathbf{z}_k|\mathbf{y}_{1:k-1}) d\mathbf{z}_k$$
$$= \mathcal{N}(\mathbf{y}_k; \hat{\mathbf{f}}_k, \hat{\mathbf{S}}_k),$$

*where $p(\mathbf{y}_k|\mathbf{z}_k) = \mathcal{N}(\mathbf{y}_k; \mathbf{z}_k, \sigma^2 \mathbf{I}_{n_s})$, $\hat{\mathbf{f}}_k := \mathbf{H}\mathbf{m}_{k|k-1}$, and $\hat{\mathbf{S}}_k := \mathbf{H}\mathbf{P}_{k|k-1}\mathbf{H}^{\top} + \sigma^2 \mathbf{I}_{n_s}$.*

See Appendix B.1 for the proof. We refer to the SSM formulation of RCGPs as ST-RCGPs, and note that the above closely relates to existing GB filtering updates (Duran-Martin et al., 2024; Reimann, 2024); but differentiates itself by using the weighted score-matching objective of Barp et al. (2019)). In contrast to what we propose in this paper, however, these previous filtering methods do *not* deal with hyperparameter optimisation and smoothing, or notions of down-weighting optimally.

Proposition 3.1 offers two key advantages over standard RCGPs: firstly, it operates with *significantly smaller* matrices of size $n_s(\nu + 1) \times n_s(\nu + 1)$ rather than $n_t n_s \times n_t n_s$ as for vanilla RCGPs; speeding up computations. Secondly, whilst vanilla RCGPs require the weight function to be fixed during inference, we can *adapt* it as filtering is performed, thereby improving inference quality. This novelty powers much of our developments, and allows us to address **Issues #1, #2,** and **#3** mentioned in Section 2. Our notation highlights this adaptivity by indexing weights with time.

The standard formulation of RCGPs arises as a particular case of Proposition 3.1 where $\mathbf{w}_k$ are specified as outlined in Section 2 and are only allowed to depend on the current observation $(\mathbf{x}_k, y_k)$. In this setting, smoothing and filtering solutions of the ST-RCGP match the RCGP batch solutions *exactly*. This emphasizes the importance of adaptivity: ST-RCGPs can fix prevailing issues of vanilla RCGP *precisely because* they allow adaptive weights, which in turn causes filtering and smoothing distributions to differ. Proposition 3.2 formalizes this result, and identifies conditions for which ST-RCGP and vanilla RCGP match.

**Proposition 3.2.** *Consider constructing a generalised posterior using $\mathcal{L}$ as defined in Equation (10), and assume that weights are chosen non-adaptively. Then, whenever the GP prior is a Gauss-Markov process, it will hold that filtering and smoothing distributions constructed sequentially as in Proposition 3.1 will be equal to those constructed in a single batch from using the vanilla RCGP form defined in Proposition 3.1 of Altamirano et al. (2024).*

See Appendix B.2 for more details and the proof.

**Weight Function** Proposition 3.1 imposes some constraints on the choice of weight function: it should be strictly positive, differentiable over its domain (this ensures that all quantities in Proposition 3.1 are well-defined). We also need the weight to be bounded from above to ensure robustness, and to avoid attributing weight to arbitrarily large $y \in \mathbb{R}$, we require that $\lim_{|y| \to \infty} w(\mathbf{x}, y) = 0$. The IMQ satisfies all these conditions, and has been well-studied and recommended in prior literature (Matsubara et al., 2022; Riabiz et al., 2022; Chen et al., 2019)—making it an ideal choice of weight function. This choice is further justified by the fact that the IMQ hyperparameters $\gamma, \beta$ and $c$ are related to concepts of robustness that we will exploit when specifying the IMQ. In particular, by selecting $\gamma, \beta$ and $c$, we want to

1. down-weight observations far from where we expect the data to be, as specified by the *center* of the data (via $\gamma$);

2. match the *rate* at which we down-weight observations with how confident we are in our estimate of the center—something that was *not* considered for RCGPs (via $c$);

3. be able to recover the STGP when there are no outliers, that is in *well-specified* settings (via $\beta$).

First, since $y \mapsto w_{\text{IMQ}}(\mathbf{x}_k, y)$ peaks at the centering function $\gamma(\mathbf{x}_k)$, we let $\gamma(\mathbf{x}_k) := \mathbb{E}_{Y_k \sim p_{\mathcal{L}}(\cdot|y_{1:k-1})}[Y_k]$, which is the mean of our GB filtering posterior predictive, thus respecting principle 1 as observations far from our prediction are down-weighted. This approach removes the vulnerability of the ST-RCGP to a poor prior mean and does not incur additional cost as filtering is part of the inference procedure of the ST-RCGP, thus resolving **Issue #1**. We also note that this estimator of the center is *robust*, since we use the GB posterior predictive, which we will see is itself robust.

Second, we observe in Figure 2 that reducing $c$ narrows $w_{\text{IMQ}}$, which in turn increases the rate $\nabla_y w_{\text{IMQ}} \propto 1/c^2$ at which observations are down-weighted. To respect principle 2, we let

$$c^2(\mathbf{x}_k) := \mathbb{E}_{Y_k \sim p_{\mathcal{L}}(\cdot|y_{1:k-1})}[(Y_k - \gamma(\mathbf{x}_k))^2],$$

which is the variance of the GB filtering posterior predictive and reflects our confidence in $\gamma$. This selection of $c$ is systematic, *adaptive*, and relates to $|y_k - \gamma(\mathbf{x}_k)|$, as it is extracted from the same distribution as $\gamma$ is. These two features are lacking in RCGPs and resolve **Issue #3**.

Finally, we set $\beta = \sigma/\sqrt{2}$, because as seen in Proposition 3.1, $\beta$ is a superfluous hyperparameter when $\sigma$ is considered, and we wish to recover the STGP in the event where $\gamma(\mathbf{x}_k) = y_k$ for $k = 1, ..., n_t$ or equivalently $\mathbf{w}_k = \beta\mathbf{1}$, accomplishing the purpose of principle 3.

In the spatio-temporal setting, $\gamma$ and $c$ are vectors $\boldsymbol{\gamma}_k := (\gamma(\mathbf{x}_{1,k}), \ldots, \gamma(\mathbf{x}_{n_s,k}))^\top$ and $\mathbf{c}_k := (c(\mathbf{x}_{1,k}), \ldots, c(\mathbf{x}_{n_s,k}))^\top$. Here, using the posterior predictive $p_{\mathcal{L}}(\mathbf{y}_k|\mathbf{y}_{1:k-1})$ for adaptivity, we take $\boldsymbol{\gamma}_k := \hat{\mathbf{f}}_k$ and $\mathbf{c}_k^2 := \text{diag}(\hat{\mathbf{S}}_k)$. In Figure 11 (presented in the appendix), we illustrate how our selection of hyperparameters improves the posterior. Although we favour the above specification of $\boldsymbol{\gamma}_k$, there are other ways to estimate the center of the data that can be appropriate in special cases (see Appendix C.11).

**Robustness** To study the robustness of ST-RCGP to outliers, we use the classical framework of Huber (1981). Consider the dataset $D = \{(\mathbf{x}_k, \mathbf{y}_k)\}_{k=1}^{n_t}$, and suppose that for some $m \in \{1, \ldots, n_t\}$ and $j \in \{1, \ldots, n_s\}$, we replace a single observation $y_{m,j}$ in the vector $\mathbf{y}_k$ by an arbitrarily large contamination $y_{m,j}^c$, resulting in the contaminated dataset $D_{m,j}^c = (D \setminus \{(\mathbf{x}_m, \mathbf{y}_m)\}) \cup \{(\mathbf{x}_m, \mathbf{y}_{m,j}^c)\}$, where $\mathbf{y}_{m,j}^c = (y_{m,1}, ..., y_{m,j}^c, ..., y_{m,n_s})$. The influence of such contamination is measured using the divergence between a posterior based on $D$ and the posterior based on $D_{m,j}^c$.

As a function of $|y_{m,j}^c - y_{m,j}|$, this divergence is called the *posterior influence function* (PIF) and has been studied in Ghosh & Basu (2016); Matsubara et al. (2022); Altamirano et al. (2023; 2024); Duran-Martin et al. (2024). For Gaussian posteriors, Altamirano et al. (2024); Duran-Martin et al. (2024) used the Kullback-Leibler (KL) divergence to

compute the PIF in closed form as follows:

$$\text{PIF}(y_{m,j}^c, D) = \text{KL}\left(p_{\mathcal{L}}(\mathbf{f}|D)\|p_{\mathcal{L}}(\mathbf{f}|D_{m,j}^c)\right).$$

We call a posterior robust if the PIF is bounded in its first input, as this implies that even as $|y_{m,j}^c - y_{m,j}| \to \infty$, the contamination's effect on the posterior is bounded. Our next results formalises the robustness of ST-RCGP.

**Proposition 3.3.** *For $w = w_{IMQ}$ with $\beta = \sigma/\sqrt{2}$, $\gamma(\mathbf{x}) = \hat{\mathbf{f}}_k$, and $c^2(\mathbf{x}_k) = diag(\hat{\mathbf{S}}_k)$ the PIF of ST-RCGP satisfies*

$$\sup_{y_{m,j}^c \in \mathbb{R}} |\text{PIF}(y_{m,j}^c, D)| < \infty,$$

*making it robust to outliers.*

See Appendix B.3 for the proof.

**Robust Hyperparameter Optimisation** The parameters $\boldsymbol{\theta}$ we need to optimize consist of the noise level $\sigma^2$ and kernel parameters such as lengthscale, amplitude and smoothness. Based on $p(\mathbf{y}_k|\mathbf{z}_k, \boldsymbol{\theta}) = \mathcal{N}(\mathbf{y}_k; \hat{\mathbf{f}}_k, \sigma^2\mathbf{I}_{n_s})$ from Section 2 and the GB posterior $p_{\mathcal{L}}(\mathbf{z}_k|\mathbf{y}_{1:k-1}, \boldsymbol{\theta}) = \mathcal{N}(\mathbf{m}_{k|k-1}(\boldsymbol{\theta}), \mathbf{P}_{k|k-1}(\boldsymbol{\theta}))$, a standard approach would be to minimise the objective

$$\varphi(\boldsymbol{\theta}) := -\sum_{k=1}^{n_t} \log p_{\mathcal{L}}(\mathbf{y}_k|\mathbf{y}_{1:k-1}, \boldsymbol{\theta}),$$

which has a closed form provided in Appendix C.3. This approach is a version of leave-one-out cross-validation, which has been proposed in Altamirano et al. (2024) to fit RCGP's hyperparameters (see Appendix C.3). Although objectives like $\varphi$ are common, they are problematic for robustness. In particular, $\varphi$ uses a form of log-likelihood loss on the posterior predictive—a loss that is well-known to be susceptible to outliers. Unsurprisingly, this makes $\varphi$ a *poor* criterion for hyperparameter optimisation—it will overfit the hyperparameters to outliers, and undo most of the robustness gains we achieved through Proposition 3.3 (see Appendix C.3). Fortunately, there is a simple and elegant way around this: weighted likelihood approaches (see e.g. Field & Smith, 1994; Windham, 1995; Dupuis & Morgenthaler, 2002; Dewaskar et al., 2023). Specifically, we define

$$\varphi_{\text{GB}}(\boldsymbol{\theta}) := -\sum_{k=1}^{n_t} \tilde{w}_k \log p_{\mathcal{L}}(\mathbf{y}_k|\mathbf{y}_{1:k-1}, \boldsymbol{\theta})$$

where $\tilde{w}_k := \tilde{w}(\mathbf{w}_k) \in \mathbb{R}$ is a function that maps the weight vector $\mathbf{w}_k$ at time $t_k$ to a type of summary indicating whether or not the observations at time $t_k$ jointly constitute an outlier along the temporal dimension of the observation process. In the absence of a spatial dimension, this value can be chosen as the weight itself so that $\tilde{w}_k := w_k$, as it reflects our belief that the corresponding point is an outlier.

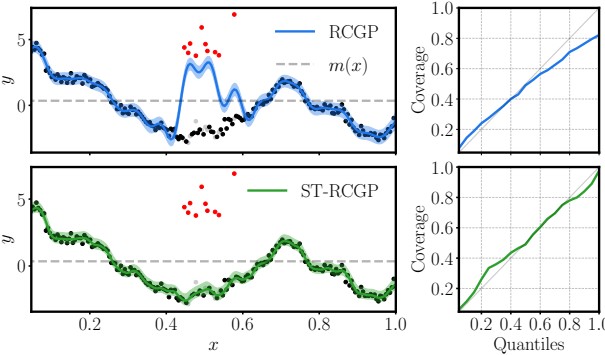

*Figure 4. Simulated Temporal Data With Focussed Outliers.* **Left:** $N = 200$ points are generated from a GP with Matérn kernel ($\nu = 3/2$), and contaminated with 5% outliers. RCGP uses $\gamma(x) = m(x) = \frac{1}{N}\sum_{i=1}^{N} y_i$ and $c = Q_N(0.95)$. While predictions are made on data with outliers, we fit the hyperparameters of RCGP on *decontaminated* data to make it as strong a competitor as possible. In contrast, ST-RCGP uses the robust hyperparameter optimisation and selection from Section 3 to fit and predict from data *with* outliers. Despite this, ST-RCGP performs much better than RCGP. **Right:** The coverage plots measure how often the uncontaminated data falls within the predictive distribution's $q$-th quantile.

In the spatio-temporal setting, however, we might instead define it as the mean, a lower quantile, or the minimum of weights across spatial locations. We refer the reader to Appendix C.3 for more details and experiments showing how $\varphi_{\mathrm{GB}}$ improves on $\varphi$ in temporal and spatio-temporal settings. On top of the better parameter estimates, we also find that the weighted approach reduces the variance in gradient computation (Wang et al., 2013), and seems to yield faster convergence. The computational cost of $\varphi_{\mathrm{GB}}$ is $\mathcal{O}(n_t n_s^3)$—the same as for inference with STGPs.

## 4. Experiments

In the remainder, we study the advantages of ST-RCGP on numerical examples. First, we investigate how ST-RCGP improves upon RCGP. Second, we explore ST-RCGP in well-specified settings without outliers. Third, we showcase its virtues on financial time series with severe outliers and its superior performance relative to competitors. Our last numerical experiment studies the robustness ST-RCGP exhibits towards spatio-temporal temperature anomalies. The code to reproduce all experiments is available at https://github.com/williamlaplante/ST-RCGP.

Throughout Section 4, we evaluate experiments based on root mean squared error $\mathrm{RMSE}(\mathbf{X}, \hat{y}) := \sqrt{\mathbb{E}_{Y \sim p_0(\cdot|\mathbf{X})} \left[(Y - \hat{y})^2\right]}$ and evaluations of the negative log predictive distribution $\mathrm{NLPD}(\mathbf{X}, \hat{y}, \hat{\sigma}) := \mathbb{E}_{Y \sim p_0(\cdot|\mathbf{X})} \left[-\log p_\phi\left(Y|\hat{y}, \hat{\sigma}^2\right)\right]$ on the test data, where $p_\phi$ is the Gaussian density with mean $\hat{y}$ and variance $\hat{\sigma}^2$ specified by the model. To capture the tradeoff between

robustness to outliers and statistical efficiency—defined in this paper as a model's ability to recover the standard GP in well-specified settings—we report the expected weighted ratio $\mathrm{EWR}(\mathbf{X}) := \mathbb{E}_{Y \sim p_0(\cdot|\mathbf{X})} \left[w(\mathbf{x}, Y)/w_{\mathrm{STGP}}(\mathbf{x}, Y)\right]$. We expand on these metrics in Appendix C.1.

Even when investigating temporal tasks without spatial dimensions, we keep the name ST-RCGP to clarify that inference proceeds (i) via the state-space representation of Proposition 3.1, and (ii) via hyperparameters chosen using the methods of Section 3—rather than those used for vanilla RCGPs in Altamirano et al. (2024).

**Fixing vanilla RCGP** In previous sections and Figure 3, we found that vanilla RCGP is vulnerable to prior mean misspecification (**Issue #1**), can produce poor uncertainty estimates (**Issue #2**) and cannot correctly select $c$ (**Issue #3**). In our first experiment, we show how ST-RCGP improves on those issues. To this end, we simulate data from a GP and compare the fit produced by both algorithms in Figure 4. The plotted predictives show that RCGP is compromised by outliers due to **Issue #1** and **Issue #3**, while the coverage plots point to unreliable uncertainty estimates induced by **Issue #2**. In contrast, using an adaptive centering and shrinking function allows ST-RCGP to produce reliable uncertainty estimates and predictions.

**ST-RCGP in Well-Specified Settings** While robust methods offer protection from contaminated data, some can only do so at the expense of statistical efficiency. Here, we show that ST-RCGP remains statistically efficient in well-specified settings and robust to outliers when required. The setup we use is described in Appendix C.13, and results are reported in Table 1. While RMSE and NLPD are comparable across methods in well-specified settings, STGP suffers from a clear drop in NLPD and RMSE when outliers are introduced. In contrast, compared to other methods, the ST-RCGP maintains the lowest RMSE and NLPD in both cases. Also, its EWR is high in well-specified settings and drops in the presence of outliers to obtain robustness. The ST-RCGP thus exhibits the properties we seek out of robust methods in well-specified settings.

*Table 1.* EWR, RMSE, and NLPD for different methods in well-specified settings and with data containing outliers.

| Outliers | Method | EWR | NLPD | RMSE |
|----------|--------|-----|------|------|
| Yes | STGP | $1.0 \pm 0.0$ | $30 \pm 2$ | $0.38 \pm 0.02$ |
| | RCGP | $0.886 \pm 0.001$ | $6.6 \pm 0.3$ | $0.21 \pm 0.01$ |
| | **ST-RCGP** | $0.863 \pm 0.002$ | $5.5 \pm 0.2$ | $0.19 \pm 0.01$ |
| No | STGP | $1.0 \pm 0.0$ | $6.5 \pm 0.2$ | $0.19 \pm 0.01$ |
| | RCGP | $0.894 \pm 0.001$ | $5.6 \pm 0.1$ | $0.19 \pm 0.01$ |
| | **ST-RCGP** | $0.903 \pm 0.001$ | $5.7 \pm 0.1$ | $0.19 \pm 0.01$ |

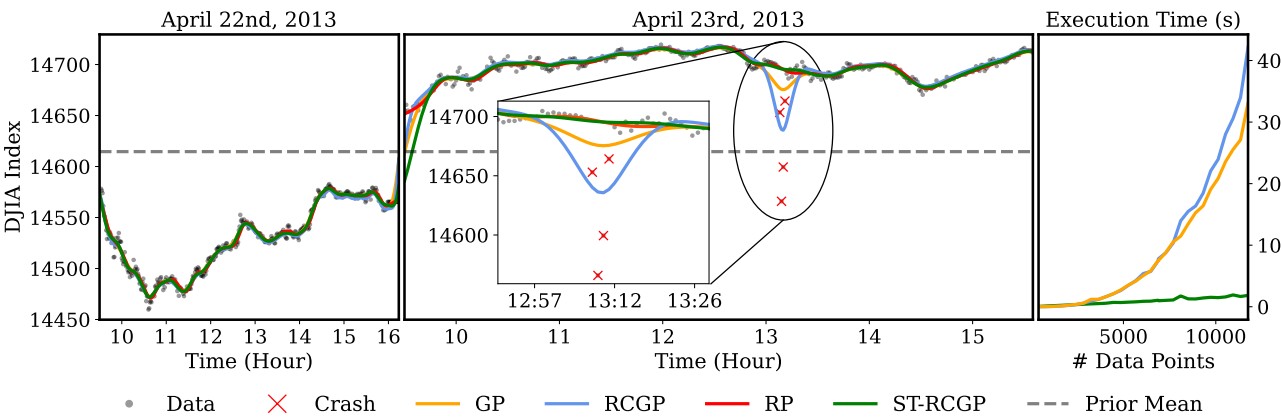

*Figure 5. Comparing Estimates of DJIA Index During Twitter Flash Crash Incident.* RCGP uses a prior mean of the data over the two days. We notice a dip in the posterior predictive of RCGP when the prior mean aligns with the crash event (red), whereas ST-RCGP and relevance pursuit algorithm (RP) from Ament et al. (2024) remain unaffected. In the right plot, we show computational time vs the number of data points. Note that RP has cost $K$ times the standard GP, where typically $K > 1$.

**Robustness During Financial Crashes** On April 23rd 2013, the Associated Press's Twitter account was hacked and posted false tweets about explosions at the White House. This led to a brief but significant market drop, including in the Dow Jones Industrial Average (DJIA), which quickly rebounded after the tweet was proven false. This data set was previously studied by Altamirano et al. (2024) and Ament et al. (2024). We plot the GP, RCGP, and ST-RCGP fits in Figure 5, as well as the robust GP via relevance pursuit (RP) fit from Ament et al. (2024). While RP behaves as desired, the plot reveals that the GP is not robust to the crash. Interestingly, the RCGP performs even *worse* since $\gamma$, which is chosen as the constant prior mean obtained by averaging two days of data, happens to be close to the outliers during the crash, implying that RCGP is centered around the outliers. This is another instance of **Issue #1**, and it is addressed by ST-RCGP. By using an adaptive centering function introduced in Section 3, ST-RCGP is centered around more reasonable values. The right-hand panel of Figure 5 also highlights that ST-RCGP leads to substantive computational gains relative to RCGP and GP due to its state-space representation from Proposition 3.1. We do not plot RP because it is implemented using a different package, and computational time is thus not comparable. But the cost of RPs is a multiple of that for GPs (Ament et al., 2024), and hence also significantly more than that of RCGPs and ST-RGCPs. It is also unclear how easily RPs could use the spatio-temporal structure to get a linear-in-time cost. For this reason, we do not explore RP beyond this experiment.

Since the flash crash dataset only contains $N = 810$ data points, we further explore the computational properties of ST-RCGP by considering a trading day of index futures data with $N = 46,800$ data points. We then synthetically induce a crash similar to that in the previous example (see

*Table 2. Performance comparison on $N = 46800$ data points with outliers.* Below, *Total (s)* denotes clock time for full inference, *1-Step (ms)* the estimated slope of a linear model fitted to execution time data for different data set sizes. Further, *RMSE* and *NLPD* use 1000 test points around the induced crash that are not outliers.

|  | Total (s) | 1-Step (ms) | RMSE | NLPD |
|---|---|---|---|---|
| STGP | $8.0 \pm 0.7$ | $0.17 \pm 0.01$ | $0.54 \pm 0.01$ | $13.9 \pm 0.3$ |
| **ST-RCGP** | $9.4 \pm 0.4$ | $0.20 \pm 0.02$ | $0.145 \pm 0.002$ | $-0.62 \pm 0.01$ |
| MEP | $29.1 \pm 0.5$ | $0.60 \pm 0.01$ | $0.15 \pm 0.01$ | $-0.57 \pm 0.03$ |
| MVI | $28.3 \pm 0.6$ | $0.61 \pm 0.01$ | $0.15 \pm 0.01$ | $-0.53 \pm 0.04$ |
| MLa | $29 \pm 2$ | $0.62 \pm 0.03$ | $0.17 \pm 0.01$ | $-0.39 \pm 0.1$ |

Appendix C.14). Naive GP implementations struggle with data of this size, and so we focus on inherently sequential methods. Table 2 summarises the results, and compares ST-RCGP against STGP, as well as several off-the-shelf inference methods for sequential GPs with Student's $t$ errors from the *BayesNewton* package (Wilkinson et al., 2023) that include Markov expectation propagation (MEP), Markov variational inference (MVI), and Markov Laplace (MLa). While STGP and ST-RCGP have similar computational cost, the robustness of ST-RCGP leads to superior performance. Conversely, ST-RCGP's performance is comparable to models using Student's $t$ errors for this problem, but its computational cost is substantively lower. This is true despite the fact that ST-RCGP is exact, while the other robust methods in Table 2 only produce approximate inferences.

**Forecasting Temperature Across the UK** Having established the performance of ST-RCGP for simpler problems, our last experiment studies its behaviour on spatio-temporal temperature data with synthetically induced outliers. In particular, we induce what are often referred to as *focussed*

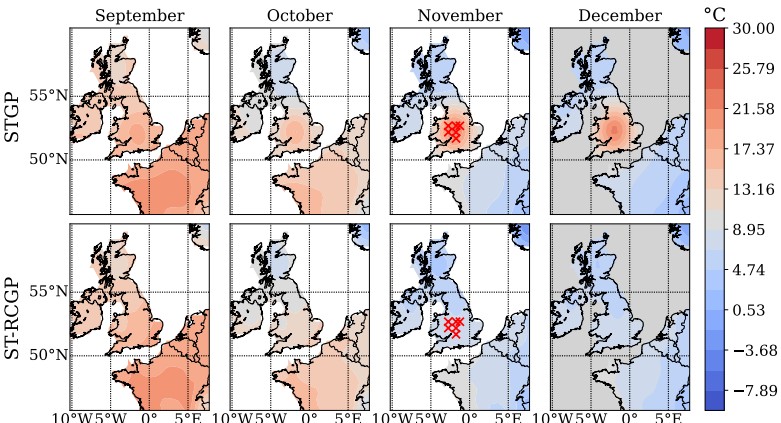

*Figure 6. Temperature fit across the UK between September and December 2023.* Focussed outliers appear in November and are shown as red crosses. We forecast the month of December. September is the last month included in parameter optimisation. We include September to show performance on previously seen data.

*Table 3. Performance on temperature data.* Models perform inference on contaminated data, but *RMSE* and *NLPD* use outlier-free data. November introduces focussed outliers, and December is a one-step forecast.

|  | STGP | | ST-RCGP | |
|---|---|---|---|---|
|  | RMSE | NLPD | RMSE | NLPD |
| May | 0.79 | 1.63 | 0.59 | 0.76 |
| Jun | 0.77 | 1.53 | 0.54 | 0.67 |
| Jul | 0.76 | 1.49 | 0.55 | 0.75 |
| Aug | 0.77 | 1.61 | 0.50 | 0.66 |
| Sep | 0.82 | 2.13 | 0.49 | 0.71 |
| Oct | 1.23 | 7.50 | 0.50 | 0.63 |
| Nov | 3.30 | 41.62 | 1.19 | 2.44 |
| Dec | 3.84 | 3.30 | 1.38 | 2.13 |

*outliers*, and which simulate the impact of rare natural phenomena that affect neighbouring weather stations. The data we use is collected by the Climate Research Unit (see Harris et al., 2020) and measures temperature from 16/01/2022 to 16/12/2023 at $n_s = 571$ locations, containing $N = 11,991$ data points in total. Hyperparameter optimisation is performed from 16/01/2022 to 30/09/2023, with later dates serving as test data. In October and November 2023, we perform in-sample predictions, and December 2023 is used for a one-month temperature forecast. Figure 6 and Table 3 display the results, and Appendix C.15 provides a coverage analysis. The two models perform similarly when there are no outliers. However, the ST-RCGP outperforms the STGP slightly because the outliers impact the STGP posterior during the months before they are introduced due to smoothing. With outliers, the STGP loses prediction accuracy (NLPD and RMSE), whereas ST-RCGP maintains consistent RMSE and NLPD over time, offering more reliable predictions at comparable computational cost.

## 5. Conclusion

We proposed ST-RCGPs based on an overhaul of the RCGP framework of Altamirano et al. (2024) that addressed some major drawbacks of vanilla RCGPs, further improved their computational efficiency and paved the way for their use in spatio-temporal problems. ST-RCGPs have the computational complexity of STGPs, but additionally provide robustness to outliers. Further, we empirically demonstrated that ST-RCGPs match the performance of the relevance pursuit algorithm from Ament et al. (2024) and of various robust GP methods from the *BayesNewton* package (Wilkinson et al., 2023) at a fraction of their computational cost and without approximation error.

Our method builds on classical STGPs, which remain vulnerable to scaling in the spatial dimension and could be addressed similarly to Hamelijnck et al. (2021) through variational approximations. Furthermore, our method may be computationally suboptimal if parallel computing is available, in which case parallel-scan algorithms (see also Särkkä & García-Fernández (2020)) could be interesting to adapt to ST-RCGPs if possible. However, we consider these investigations to be outside the scope of this paper. An entirely different but equally relevant endeavour for future research is to explore the use of non-Gaussian likelihoods within the RCGP framework. This would extend its use beyond the standard regression setting, and allow its application to classification problems, count data, and various bounded data domains.

## Acknowledgements

WL was supported by UCL's Center for Doctoral Training in Data-Intensive Science and by the Alan Turing Institute. MA was supported by a Bloomberg Data Science PhD fellowship. FXB and JK were supported by the EPSRC grant EP/Y011805/1, and JK was additionally supported by EP/W005859/1.

## Impact Statement

This paper presents work whose goal is to advance the field of Machine Learning. There are many potential societal consequences of our work, none which we feel must be specifically highlighted here.

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

# Supplementary Material

The Appendix is structured as follows: We first introduce notation. Then, in Appendix A, we derive the loss function used in the main paper. Next, in Appendix B, we provide the proofs of all our theoretical results. Finally, in Appendix C, we provide additional details on our numerical experiments, as well as further experiments that complement those in the main text.

## Notation

Suppose we have a vector $\mathbf{v} = (v_1, v_2, ..., v_N)^\top \in \mathbb{R}^N$, where $v_i : \mathbb{R} \to \mathbb{R}$, and a matrix $\mathbf{M} \in \mathbb{R}^{N \times N}$. Then,

- $\text{diag}(\mathbf{v}) := \begin{pmatrix} v_1 & \cdots & 0 & 0 \\ \vdots & \ddots & \vdots & \vdots \\ 0 & \cdots & v_{N-1} & 0 \\ 0 & \cdots & 0 & v_N \end{pmatrix}$ and $\text{diag}(\mathbf{M}) := (M_{11}, M_{22}, \ldots, M_{NN})^\top$ where $M_{ij} := (\mathbf{M})_{ij}$.

- $\mathbf{v}^2 := (v_1^2, ..., v_N^2)^\top$

- $\log(\mathbf{v}) := (\log(v_1), \ldots, \log(v_N))^\top$

- $\nabla_y \mathbf{v} = \left( \frac{\partial}{\partial y} v_1, \ldots, \frac{\partial}{\partial y} v_N \right)^\top$, where $\frac{\partial}{\partial y} v_i$ is the partial derivative of $v_i$ with respect to $y$.

- $\mathbf{v} \odot \mathbf{w} := (v_1 w_1, ..., v_N w_N)^\top$ for $\mathbf{w} \in \mathbb{R}^N$.

We now remind the reader of the dimensionality of matrices and vectors crucial to the main algorithm (Proposition 3.1) of this paper:

- $\mathbf{A}_{k-1}, \mathbf{\Sigma}_{k-1} \in \mathbb{R}^{n_s(\nu+1) \times n_s(\nu+1)}$ are the transition matrices from Equation (4)

- $\mathbf{H} \in \mathbb{R}^{n_s \times n_s(\nu+1)}$ is the measurement matrix from Equation (4)

- $\mathbf{y}_k, \mathbf{w}_k, \hat{\mathbf{f}}_k^{\text{GB}} \in \mathbb{R}^{n_s}$ are the observations, weights and filtering predictives from Proposition 3.1.

- $\mathbf{P}_{k|k}^{\text{GB}} \in \mathbb{R}^{n_s(\nu+1) \times n_s(\nu+1)}$ is the covariance matrix from Proposition 3.1

- $\mathbf{m}_{k|k}^{\text{GB}} \in \mathbb{R}^{n_s(\nu+1)}$ is the mean from Proposition 3.1

- $\mathbf{K}_k^{\text{GB}} \in \mathbb{R}^{n_s(\nu+1) \times n_s}$ is the Kalman gain matrix from Proposition 3.1

- $\mathbf{J}_{\mathbf{w}_k} \in \mathbb{R}^{n_s \times n_s}$ is the weight-based matrix from Proposition 3.1.

Finally, when we use $p(\cdot) = \mathcal{N}(\cdot \ ; \mu, \Sigma)$, we are referring to the density of the normal distribution with mean $\mu$ and covariance $\Sigma$, and when we use $\mathcal{N}(\mu, \Sigma)$, we are referring to the distribution.

## A. Score-Matching & Loss Function

This section provides the derivation of the loss function used in the main paper based on the weighted Fisher divergence.

We define the weighted Fisher divergence at a fixed time $t_k$. Let $f_k(.) = f(., t_k)$ the spatial model at time $t_k$. Let $p_k(y|s) = p(y|s, t_k)$ be the density of the true data-generating process at time $t_k$, and $p_{f_k}(y|s) = p(y|f_k(s))$ be the density of our model at time $t_k$.

The *weighted Fisher divergence* for regression between the model and data-generating process at time $t_k$ depends on the corresponding scores $s_{f_k}(y|s) = \nabla_y \log p_{f_k}(y|s)$ and $s_k(y|s) = \nabla_y \log p_k(y|s)$ and is given by (Barp et al., 2019; Xu et al., 2022; Altamirano et al., 2024):

$$\mathcal{D}_k := \mathbb{E}_{S \sim p_{k,s}} \left[ \mathbb{E}_{Y \sim p_k(.|S)} \left[ \| w_k(S,Y) \left( s_k(Y|S) - s_{f_k}(Y|S) \right) \|_2^2 \right] \right]$$

$$\stackrel{+C}{=} \mathbb{E}_{S \sim p_{k,s}} \left[ \mathbb{E}_{Y \sim p_k(.|S)} \left[ \| w_k(S,Y) s_{f_k}(Y|S) \|_2^2 + 2 \nabla_y \cdot \left( w_k(S,Y)^2 s_{f_k}(Y|S) \right) \right] \right],$$

where $p_{k,s} = p(s|t_k)$ is the marginal, $w_k(s,y) = w((s,t_k),y)$, and the equality—which crucially does not depend on $s_k$ anymore—holds up to an additive constant $C$ not depending on $f_k$.

Now, consider a dataset such that $\mathbf{x}_k = (\mathbf{x}_{k,1}, ..., \mathbf{x}_{k,n_s})^\top$, for $\mathbf{x}_{k,j} = (\mathbf{s}_j, t_k) \in \mathcal{X} = \mathcal{S} \times \mathcal{T}$, $\mathbf{y}_k = (y_{k,1}, ..., y_{k,n_s})^\top$, and let $\mathbf{f}_k := \mathbf{f}(\mathbf{x}_k) = (f(\mathbf{x}_{k,1}), ..., f(\mathbf{x}_{k,n_s}))^\top$. Moreover, let $\mathbf{z}_k$ be related to $\mathbf{f}_k$ as $\mathbf{f}_k := \mathbf{H}\mathbf{z}_k$. The empirical loss we obtain for filtering is then

$$\mathcal{L}(\mathbf{x}_k, \mathbf{y}_k, \mathbf{z}_k) = \frac{1}{n_s} \sum_{j=1}^{n_s} \| w_{k,j} s_{f,k,j} \|_2^2 + 2 \nabla_y \cdot \left( (w_{k,j})^2 s_{f,k,j} \right), \tag{11}$$

where $w_{k,j} = w(\mathbf{x}_{k,j}, y_{k,j})$, and $s_{f,k,j} = s_{f_k}(y_{k,j})$.

## B. Proofs of Theoretical Results

### B.1. Proof of Proposition 3.1

In the following, we derive the generalised Bayes filtering posterior distribution when the loss function is quadratic. We then show that the weighted score matching loss with a Gaussian model yields a quadratic loss. Finally, we provide the derivation of the GB predictive.

Let us assume a quadratic loss in $\mathbf{z}_k$ or equivalently in $\mathbf{f}_k := \mathbf{H}\mathbf{z}_k$ of the form:

$$\mathcal{L}(\mathbf{x}_k, \mathbf{y}_k, \mathbf{z}_k) = \frac{1}{2n_s} \left( \mathbf{f}_k^\top \mathbf{R}_k^{-1} \mathbf{f}_k - \mathbf{f}_k^\top \mathbf{v}_k + C_k \right),$$

where $R_k \in \mathbb{R}^{n_s \times n_s}$ is an invertible matrix, $v_k \in \mathbb{R}^{n_s}$, and $C_k \in \mathbb{R}$. The GB filtering update equations are then

$$p_\mathcal{L}(\mathbf{z}_k|\mathbf{y}_{1:k}) \propto p(\mathbf{z}_k|\mathbf{y}_{1:k-1}) \exp(-n_s \mathcal{L}(\mathbf{x}_k, \mathbf{y}_k, \mathbf{z}_k))$$

$$\propto \exp\left( -\frac{1}{2} (\mathbf{z}_k - \mathbf{m}_{k|k-1})^\top \mathbf{P}_{k|k-1}^{-1} (\mathbf{z}_k - \mathbf{m}_{k|k-1}) \right) \exp\left( -\frac{1}{2} \left( \mathbf{f}_k^\top \mathbf{R}_k^{-1} \mathbf{f}_k - 2\mathbf{f}_k^\top \mathbf{v}_k \right) \right) \tag{12}$$

$$\propto \exp\left( -\frac{1}{2} \left( \mathbf{z}_k^\top (\mathbf{P}_{k|k-1}^{-1} + \mathbf{H}^\top \mathbf{R}_k^{-1} \mathbf{H}) \mathbf{z}_k - 2\mathbf{z}_k^\top (\mathbf{P}_{k|k-1}^{-1} \mathbf{m}_{k|k-1} + \mathbf{H}^\top \mathbf{v}_k) \right) \right),$$

which implies that the mean $\mathbf{m}_{k|k}^{\text{GB}}$ and covariance $\mathbf{P}_{k|k}^{\text{GB}}$ of the GB posterior $p_\mathcal{L}$ are:

$$\mathbf{P}_{k|k}^{\text{GB}} = \left( \mathbf{P}_{k|k-1}^{-1} + \mathbf{H}^\top \mathbf{R}_k^{-1} \mathbf{H} \right)^{-1}$$

$$\mathbf{m}_{k|k}^{\text{GB}} = \mathbf{P}_{k|k}^{\text{GB}} \left( \mathbf{P}_{k|k-1}^{-1} \mathbf{m}_{k|k-1} + \mathbf{H}^\top \mathbf{v}_k \right). \tag{13}$$

As with the typical Kalman filter, those equations can be written in the form

$$\mathbf{P}_{k|k}^{\text{GB}} = \left( \mathbf{P}_{k|k-1}^{-1} + \mathbf{H}^\top \mathbf{R}_k^{-1} \mathbf{H} \right)^{-1}$$

$$\mathbf{K}_k^{\text{GB}} = \mathbf{P}_{k|k}^{\text{GB}} \mathbf{H}^\top \mathbf{R}_k^{-1} \tag{14}$$

$$\mathbf{m}_{k|k}^{\text{GB}} = \mathbf{m}_{k|k-1} + \mathbf{K}_k^{\text{GB}} (\mathbf{R}_k \mathbf{v}_k - \mathbf{H}\mathbf{m}_{k|k-1}),$$

where $\mathbf{K}_k^{\text{GB}}$ is the Kalman gain matrix. The typical Kalman filter equations—those used for STGPs—are recovered when $\mathbf{v}_k := \mathbf{R}_k^{-1} \mathbf{y}_k$ and $\mathbf{R}_k^{-1} := \sigma^{-2} \mathbf{I}_{n_s}$.

**Weighted score matching and Gaussian likelihood.** We now show that the loss function defined in Appendix A and Gaussian likelihood lead to a quadratic loss in $\mathbf{f}_k$. We have a Gaussian likelihood, which gives a score function of the form:

$$p(\mathbf{y}_k|\mathbf{z}_k, \mathbf{x}_k) \propto \exp\left(-\frac{1}{2}(\mathbf{y}_k - \mathbf{f}_k)^\top \sigma^{-2}\mathbf{I}_{n_s}(\mathbf{y}_k - \mathbf{f}_k)\right) \implies \mathbf{s}_{f,k} = \nabla_y \log p(\mathbf{y}_k|\mathbf{f}_k) = (\mathbf{f}_k - \mathbf{y}_k)^\top \sigma^{-2}\mathbf{I}_{n_s}, \quad (15)$$

where $\mathbf{f}_k := \mathbf{H}\mathbf{z}_k$, $\mathbf{z}_k = (z_{k,1}, ..., z_{k,n_s})^\top$, $\mathbf{y}_k = (y_{k,1}, ..., y_{k,n_s})^\top$ and $\mathbf{s}_{f,k} = (s_{f,k,1}, ..., s_{f,k,n_s})^\top$. Then, the loss is given by

$$
\begin{aligned}
\mathcal{L}(\mathbf{x}_k, \mathbf{y}_k, \mathbf{z}_k) =& \frac{1}{n_s}\sum_{j=1}^{n_s}(w_{k,j}s_{f,k,j})^2 + 2\nabla_y\left(w_{k,j}^2 s_{f,k,j}\right) \\
=& \frac{1}{n_s}\sum_{j=1}^{n_s}\left(w_{k,j}\frac{(f_{k,j} - y_{k,j})}{\sigma^2}\right)^2 + 2\nabla_y\left(w_{k,j}^2\frac{(f_{k,j} - y_{k,j})}{\sigma^2}\right) \\
=& \frac{1}{n_s}\sum_{j=1}^{n_s}\frac{1}{\sigma^4}w_{k,j}^2 f_{k,j}^2 - \frac{2}{\sigma^4}f_{k,j}y_{k,j}w_{k,j}^2 + \frac{1}{\sigma^4}w_{k,j}^2 y_{k,j}^2 + \frac{2}{\sigma^2}\nabla_y(w_{k,j}^2 f_{k,j}) - \frac{2}{\sigma^2}\nabla_y(w_{k,j}^2 y_{k,j})
\end{aligned}
$$

Now let us group all the elements that do not depend on $f$ and call it $C(y_{k,j})$

$$
\begin{aligned}
\mathcal{L}(\mathbf{x}_k, \mathbf{y}_k, \mathbf{z}_k) =& \frac{1}{n_s}\sum_{j=1}^{n_s}\frac{1}{\sigma^4}w_{k,j}^2 f_{k,j}^2 - \frac{2}{\sigma^4}f_{k,j}y_{k,j}w_{k,j}^2 + \frac{2}{\sigma^2}\nabla_y(w_{k,j}^2 f_{k,j}) + C(y_{k,j}) \\
=& \frac{1}{n_s}\sum_{j=1}^{n_s}\frac{1}{\sigma^4}w_{k,j}^2 f_{k,j}^2 - \frac{2}{\sigma^4}f_{k,j}y_{k,j}w_{k,j}^2 + \frac{4}{\sigma^2}w_{k,j}f_{k,j}\nabla_y w_{k,j} + C(y_{k,j}) \\
=& \frac{1}{n_s}\sum_{j=1}^{n_s}\frac{1}{\sigma^4}w_{k,j}^2 f_{k,j}^2 - \frac{2}{\sigma^4}w_{k,j}^2\left(y_{k,j} - 2\sigma^2(w_{k,j})^{-1}\nabla_y w_{k,j}\right)f_{k,j} + C(y_{k,j})
\end{aligned}
$$

Next, we rewrite the loss in terms of vectors and matrices as follows:

$$\mathcal{L}(\mathbf{x}_k, \mathbf{y}_k, \mathbf{z}_k) = \frac{1}{2n_s}\left(\frac{2}{\sigma^4}\mathbf{f}_k^\top \mathrm{diag}(\mathbf{w}_k^2)\mathbf{f}_k - \frac{4}{\sigma^4}\mathbf{f}_k^\top\left(\mathrm{diag}(\mathbf{w}_k^2)\mathbf{y}_k - 2\sigma^2\mathrm{diag}(\mathbf{w}_k)\nabla_y\mathbf{w}_k + C(\mathbf{y}_k)\right)\right),$$

where $C(\mathbf{y}_k) = \sum_{j=1}^{n_s}C(y_{k,j})$, $\mathrm{diag}(\mathbf{w}_k^2) \in \mathbb{R}^{n_s \times n_s}$ is the diagonal matrix of the vector $\mathbf{w}_k^2 = \mathbf{w}_k \odot \mathbf{w}_k$ for $\odot$ the element-wise multiplication operator, and $\nabla_y\mathbf{w}_k = (\nabla_y w_{k,1}, ..., \nabla_y w_{k,n_s})^\top$. This leads to

$$
\begin{aligned}
\mathbf{R}^{-1}(\mathbf{y}_k; \mathbf{w}_k) :=& \frac{2}{\sigma^4}\mathrm{diag}(\mathbf{w}_k^2) \\
\mathbf{v}_k(\mathbf{y}_k; \mathbf{w}_k) :=& \frac{2}{\sigma^4}\left(\mathrm{diag}(\mathbf{w}_k^2)\mathbf{y}_k - 2\sigma^2\mathrm{diag}(\mathbf{w}_k)\nabla_y\mathbf{w}_k\right).
\end{aligned}
\quad (16)
$$

**GP predictive** The GB predictive can be written as:

$$p_{\mathcal{L}}(\mathbf{y}_k|\mathbf{y}_{1:k-1}) = \int p(\mathbf{y}_k|\mathbf{z}_k)p_{\mathcal{L}}(\mathbf{z}_k|\mathbf{y}_{1:k-1})d\mathbf{z}_k$$

where $p(\mathbf{y}_k|\mathbf{z}_k) = \mathcal{N}(\mathbf{y}_k; \mathbf{H}\mathbf{z}_k, \sigma^2\mathbf{I}_{n_s})$ is the likelihood and $p_{\mathcal{L}}(\mathbf{z}_k|\mathbf{y}_{1:k}) = \mathcal{N}(\mathbf{z}_k; \mathbf{m}_{k|k-1}, \mathbf{P}_{k|k-1})$ is the predict step defined in Proposition 3.1. Since both densities are Gaussian, this integral is known and the solution is also a Gaussian of the form:

$$p_{\mathcal{L}}(\mathbf{y}_k|\mathbf{y}_{1:k-1}) = \int \mathcal{N}(\mathbf{y}_k; \mathbf{H}\mathbf{z}_k, \sigma^2\mathbf{I}_{n_s})\mathcal{N}(\mathbf{z}_k; \mathbf{m}_{k|k-1}, \mathbf{P}_{k|k-1})d\mathbf{z}_k = \mathcal{N}(\mathbf{y}_k; \mathbf{H}\mathbf{m}_{k|k-1}, \mathbf{H}\mathbf{P}_{k|k-1}\mathbf{H}^\top + \sigma^2\mathbf{I}_{n_s})$$

### B.2. Proof that ST-RCGP reproduces RCGP for Proposition 3.2

We first define the necessary quantities. Let the posterior density of RCGP be

$$p_{\text{RCGP}}(\mathbf{f}_{1:k}|\mathbf{x}_{1:k}, \mathbf{y}_{1:k}) \propto p(\mathbf{f}_{1:k}|\mathbf{x}_{1:k}) \exp(-k n_s \mathcal{L}_{\text{RCGP}}(\mathbf{x}_{1:k}, \mathbf{y}_{1:k}, \mathbf{f}_{1:k})),$$

where $p(\mathbf{f}_{1:k}|\mathbf{x}_{1:k})$ is the prior density, and

$$\mathcal{L}_{\text{RCGP}}(\mathbf{x}_{1:k}, \mathbf{y}_{1:k}, \mathbf{f}_{1:k}) := \frac{1}{k n_s} \sum_{i=1}^{k} \sum_{j=1}^{n_s} (w(\mathbf{x}_{i,j}, y_{i,j}) s_{f_i}(y_{i,j}))^2 + 2 \nabla_y (w^2(\mathbf{x}_{i,j}, y_{i,j}) s_{f_i}(y_{i,j})), \qquad (17)$$

where $i$ indexes time, $j$ indexes spatial coordinates, and $k \in \{1, ..., n_t\}$. Note that we now have dependence on $\mathbf{f}_k$ instead of $\mathbf{z}_k$ for simplicity of notation; however the two definitions are equivalent since $\mathbf{f}_k = \mathbf{H}\mathbf{z}_k$.

For fixed hyperparameters of $w$ (for example, $c, \beta$ for $w_{\text{IMQ}}$), the loss $\mathcal{L}$ from Proposition 3.1 relates to $\mathcal{L}_{\text{RCGP}}$ as

$$\mathcal{L}_{\text{RCGP}}(\mathbf{x}_{1:k}, \mathbf{y}_{1:k}, \mathbf{f}_{1:k}) = \frac{1}{k} \sum_{i=1}^{k} \mathcal{L}(\mathbf{x}_i, \mathbf{y}_i, \mathbf{f}_i), \qquad (18)$$

which implies that the loss $\mathcal{L}_{\text{RCGP}}$ is *summable* (can be broken down into a summation over $i = 1, \ldots, k$). Moreover, let the posterior filtering distribution of ST-RCGP on $\mathbf{f}_k$ be

$$p_{\text{ST-RCGP}}(\mathbf{f}_k|\mathbf{x}_{1:k}, \mathbf{y}_{1:k}) \propto p_{\text{ST-RCGP}}(\mathbf{f}_k|\mathbf{x}_{1:k}, \mathbf{y}_{1:k-1}) \exp(-n_s \mathcal{L}(\mathbf{x}_k, \mathbf{y}_k, \mathbf{f}_k))$$

$$p_{\text{ST-RCGP}}(\mathbf{f}_k|\mathbf{x}_{1:k}, \mathbf{y}_{1:k-1}) = \int p(\mathbf{f}_k|\mathbf{x}_{1:k}, \mathbf{f}_{k-1}) p_{\text{ST-RCGP}}(\mathbf{f}_{k-1}|\mathbf{x}_{1:k-1}, \mathbf{y}_{1:k-1}) d\mathbf{f}_{k-1},$$

where the loss $\mathcal{L}$ is specified as in Proposition 3.1.

The above definitions are such that the prior density is the same for $p_{\text{RCGP}}$ and $p_{\text{ST-RCGP}}$, and the loss $\mathcal{L}$, which is summable, is specified identically. If we further impose on the Gaussian process to be Markovian (the so-called Gauss-Markov process), which is a requirement for the Gaussian process to be expressed as a state-space model, these assumptions will allow us to postulate the following Proposition 3.2:

**Proposition B.1.** *Suppose that*

  *(i) the prior distribution $p(\mathbf{f}_{1:k}|\mathbf{x}_{1:k})$ is identical for $p_{RCGP}$ and $p_{ST\text{-}RCGP}$;*

  *(ii) the loss function $\mathcal{L}$ is defined as in Proposition 3.1 and $\mathcal{L}_{RCGP}(\mathbf{x}_{1:k}, \mathbf{y}_{1:k}, \mathbf{f}_{1:k})$ as in Equation (17), with weights that do not depend on past observations;*

  *(iii) the GP prior $f \sim \mathcal{N}(m, \kappa)$ is a Gauss-Markov process and can be expressed as a state-space model.*

*Then,*

  *1. $p_{RCGP}(\mathbf{f}_k|\mathbf{x}_{1:k}, \mathbf{y}_{1:k}) = p_{ST\text{-}RCGP}(\mathbf{f}_k|\mathbf{x}_{1:k}, \mathbf{y}_{1:k})$ (filtering posteriors are equal)*

  *2. $p_{RCGP}(\mathbf{f}_k|\mathbf{x}_{1:n_t}, \mathbf{y}_{1:n_t}) = p_{ST\text{-}RCGP}(\mathbf{f}_k|\mathbf{x}_{1:n_t}, \mathbf{y}_{1:n_t})$ (smoothing solutions are equal).*

*Proof.* The proof is divided into two steps: the first to tackle claim 1., which is that the filtering posteriors are equal, and the second to claim that the smoothing distributions are equal (claim 2.). In both cases, we proceed inductively, with a base case and an inductive step.

We first want to show that $p_{\text{RCGP}}(\mathbf{f}_k|\mathbf{x}_{1:k}, \mathbf{y}_{1:k}) = p_{\text{ST-RCGP}}(\mathbf{f}_k|\mathbf{x}_{1:k}, \mathbf{y}_{1:k})$.

**Step 1:** $p_{\text{RCGP}}(\mathbf{f}_k|\mathbf{x}_{1:k}, \mathbf{y}_{1:k}) = p_{\text{ST-RCGP}}(\mathbf{f}_k|\mathbf{x}_{1:k}, \mathbf{y}_{1:k})$ (claim 1.).

Base Case:

Consider $k = 1$. Then from the RCGP equations in Altamirano et al. (2024) (Proposition 3.1) and in our own Proposition 3.1, the RCGP posterior and the ST-RCGP posterior are:

$$p_{\text{RCGP}}(\mathbf{f}_1|\mathbf{x}_1, \mathbf{y}_1) = \mathcal{N}(\mathbf{f}_1; \mu_{\text{RCGP}}, \Sigma_{\text{RCGP}})$$

$$p_{\text{ST-RCGP}}(\mathbf{f}_1|\mathbf{x}_1, \mathbf{y}_1) = \mathcal{N}(\mathbf{f}_1; \mu_{\text{ST-RCGP}}, \Sigma_{\text{ST-RCGP}}),$$

where

$$\mu_{\text{RCGP}} = \mathbf{m}_1 + \mathbf{K}_1 \left(\mathbf{K}_1 + \sigma^2 \mathbf{J}_{w_1}\right)^{-1} \left(\mathbf{y}_1 - \mathbf{m}_{w_1}\right)$$
$$\Sigma_{\text{RCGP}} = \mathbf{K}_1 \left(\mathbf{K}_1 + \sigma^2 \mathbf{J}_{w_1}\right)^{-1} \sigma^2 \mathbf{J}_{w_1} \tag{19}$$

and

$$\mu_{\text{ST-RCGP}} = \mathbf{H}\mathbf{m}_{1|0} + \mathbf{H}\mathbf{P}_{1|1}^{\text{GB}}\mathbf{H}^\top \left(\sigma^2 \mathbf{J}_{w_1}\right)^{-1} \left(\mathbf{y}_1 - \hat{\mathbf{f}}_{w_1}\right)$$
$$\Sigma_{\text{ST-RCGP}} = \mathbf{H} \left(\left(\mathbf{P}_{1|0}^{\text{GB}}\right)^{-1} + \mathbf{H}^\top \sigma^{-2} \mathbf{J}_{w_1}^{-1} \mathbf{H}\right)^{-1} \mathbf{H}^\top \tag{20}$$

Then, we expand $\Sigma_{\text{RCGP}}$ as follows:

$$\begin{aligned}
\Sigma_{\text{RCGP}} &= \mathbf{K}_1 \left(\mathbf{K}_1 + \sigma^2 \mathbf{J}_{w_1}\right)^{-1} \sigma^2 \mathbf{J}_{w_1} \\
&= \mathbf{K}_1 - \mathbf{K}_1 \left(\sigma^2 \mathbf{J}_{w_1}\right)^{-1} \left(\mathbf{K}_1^{-1} + \left(\sigma^2 \mathbf{J}_{w_1}\right)^{-1}\right)^{-1} \\
&= \mathbf{K}_1 - \mathbf{K}_1 \left(\mathbf{K}_1 + \sigma^2 \mathbf{J}_{w_1}\right)^{-1} \mathbf{K}_1,
\end{aligned}$$

where in the first line, we apply the Woodbury identity (Chapter 2.1.3 Golub & Van Loan, 2013), and in the second, we use the fact that for two invertible matrices $A, B$, we have $(A^{-1} + B^{-1})^{-1} = A(A + B)^{-1}B$. Moreover, since $\mathbf{f}_k = \mathbf{H}\mathbf{z}_k$, $\Sigma_{\text{ST-RCGP}} = \mathbf{H}\mathbf{P}_{1|1}^{\text{GB}}\mathbf{H}^\top$, so that

$$\begin{aligned}
\Sigma_{\text{ST-RCGP}} &= \mathbf{H} \left(\left(\mathbf{P}_{1|0}^{\text{GB}}\right)^{-1} + \mathbf{H}^\top \sigma^{-2} \mathbf{J}_{w_1}^{-1} \mathbf{H}\right)^{-1} \mathbf{H}^\top \\
&= \mathbf{H}\mathbf{P}_{1|0}^{\text{GB}}\mathbf{H}^\top - \mathbf{H}\mathbf{P}_{1|0}^{\text{GB}}\mathbf{H}^\top \left(\mathbf{H}\mathbf{P}_{1|0}^{\text{GB}}\mathbf{H}^\top + \sigma^2 \mathbf{J}_{w_1}\right)^{-1} \mathbf{H}\mathbf{P}_{1|0}^{\text{GB}}\mathbf{H}^\top,
\end{aligned}$$

where we again use the Woodbury identity. But, both matrices $\mathbf{H}\mathbf{P}_{1|0}^{\text{GB}}\mathbf{H}^\top$ and $\mathbf{K}_1$ represent the covariance matrix of $p(\mathbf{f}_1|\mathbf{x}_1)$, and thus must be equal by the assumption of identical prior (assumption (i)). Therefore, substituting $\mathbf{K}_1 = \mathbf{H}\mathbf{P}_{1|0}^{\text{GB}}\mathbf{H}^\top$ in either of the expression for $\Sigma_{\text{RCGP}}$ or $\Sigma_{\text{ST-RCGP}}$ yields $\Sigma_{\text{RCGP}} = \Sigma_{\text{ST-RCGP}}$.

Now, we have

$$\mu_{\text{RCGP}} = \mathbf{m}_1 + \mathbf{K}_1 \left(\mathbf{K}_1 + \sigma^2 \mathbf{J}_{w_1}\right)^{-1} \left(\mathbf{y}_1 - \mathbf{m}_{w_1}\right) = \mathbf{m}_1 + \Sigma_{\text{RCGP}} \left(\sigma^2 \mathbf{J}_{w_1}\right)^{-1} \left(\mathbf{y}_1 - \mathbf{m}_{w_1}\right)$$

where in the second equality, we substitute $\Sigma_{\text{RCGP}}$ from Equation (19). Since $\mathbf{H}\mathbf{P}_{1|1}\mathbf{H}^\top$ is the covariance of $p_{\text{ST-RCGP}}(\mathbf{f}_1|\mathbf{x}_1, \mathbf{y}_1)$, then, $\mathbf{H}\mathbf{P}_{1|1}\mathbf{H}^\top = \Sigma_{\text{ST-RCGP}} = \Sigma_{\text{RCGP}}$ (last equality holds from previous step). Moreover, $\mathbf{H}\mathbf{m}_{1|0} = \hat{\mathbf{f}}_1$ and $\mathbf{m}_1$ are both the mean of $p(\mathbf{f}_1)$. Since the priors are identical, we then have $\mathbf{H}\mathbf{m}_{1|0} = \hat{\mathbf{f}}_1 = \mathbf{m}_1$, and by extension $\mathbf{m}_{w_1} = \hat{\mathbf{f}}_{w_1}$ since the two distributions are defined with the same loss function (and thus have identical weights). Then, as before, substituting these quantities yields $\mu_{\text{RCGP}} = \mu_{\text{ST-RCGP}}$, concluding the base case.

Inductive Step:

Suppose that $p_{\text{ST-RCGP}}(\mathbf{f}_{k-1}|\mathbf{x}_{1:k-1}, \mathbf{y}_{1:k-1}) = p_{\text{RCGP}}(\mathbf{f}_{k-1}|\mathbf{x}_{1:k-1}, \mathbf{y}_{1:k-1})$ for some $k > 2$. Then,

$$\begin{aligned}
p_{\text{RCGP}}(\mathbf{f}_k|\mathbf{x}_{1:k}, \mathbf{y}_{1:k}) &= \int p_{\text{RCGP}}(\mathbf{f}_{1:k}|\mathbf{x}_{1:k}, \mathbf{y}_{1:k}) d\mathbf{f}_{1:k-1} \\
&\propto \int \exp(-\mathcal{L}(\mathbf{x}_{1:k}, \mathbf{y}_{1:k}, \mathbf{f}_{1:k})) p(\mathbf{f}_{1:k}|\mathbf{x}_{1:k}) d\mathbf{f}_{1:k-1}
\end{aligned} \tag{21}$$

But $p(\mathbf{f}_{1:k}|\mathbf{x}_{1:k}) = p(\mathbf{f}_k|\mathbf{f}_{1:k-1}, \mathbf{x}_{1:k})p(\mathbf{f}_{1:k-1}|\mathbf{x}_{1:k}) = p(\mathbf{f}_k|\mathbf{f}_{k-1}, \mathbf{x}_{1:k})p(\mathbf{f}_{1:k-1}|\mathbf{x}_{1:k})$ by the Markov assumption on $\mathbf{f}$ (assumption (iii)). That is, $\mathbf{f}_k$ is conditionally independent of $\mathbf{f}_{1:k-2}$ given $\mathbf{f}_{k-1}$.

Also, $\exp(-\mathcal{L}(\mathbf{x}_{1:k}, \mathbf{y}_{1:k}, \mathbf{f}_{1:k})) = \exp(-\mathcal{L}(\mathbf{x}_k, \mathbf{y}_k, \mathbf{f}_k))\exp(-\mathcal{L}(\mathbf{x}_{1:k-1}, \mathbf{y}_{1:k-1}, \mathbf{f}_{1:k-1}))$ since the loss is summable. Therefore,

$$p_{\text{RCGP}}(\mathbf{f}_k|\mathbf{x}_{1:k}, \mathbf{y}_{1:k}) \propto \exp(-\mathcal{L}(\mathbf{x}_k, \mathbf{y}_k, \mathbf{f}_k)) \int p(\mathbf{f}_k|\mathbf{f}_{k-1}, \mathbf{x}_{1:k})p(\mathbf{f}_{1:k-1}|\mathbf{x}_{1:k})\exp(-\mathcal{L}(\mathbf{x}_{1:k-1}, \mathbf{y}_{1:k-1}, \mathbf{f}_{1:k-1}))d\mathbf{f}_{1:k-1},$$

where we substitute $p(\mathbf{f}_{1:k}|\mathbf{x}_{1:k}) = p(\mathbf{f}_k|\mathbf{f}_{k-1}, \mathbf{x}_{1:k})p(\mathbf{f}_{1:k-1}|\mathbf{x}_{1:k})$ and $\exp(-\mathcal{L}(\mathbf{x}_{1:k}, \mathbf{y}_{1:k}, \mathbf{f}_{1:k})) = \exp(-\mathcal{L}(\mathbf{x}_k, \mathbf{y}_k; \mathbf{f}_k))\exp(-\mathcal{L}(\mathbf{x}_{1:k-1}, \mathbf{y}_{1:k-1}, \mathbf{f}_{1:k-1}))$. Now, using the definition of the $p_{\text{RCGP}}$ for $k-1$:

$$p_{\text{RCGP}}(\mathbf{f}_k|\mathbf{x}_{1:k}, \mathbf{y}_{1:k}) \propto \exp(-\mathcal{L}(\mathbf{x}_k, \mathbf{y}_k, \mathbf{f}_k)) \int p(\mathbf{f}_k|\mathbf{f}_{k-1}, \mathbf{x}_{1:k})p_{\text{RCGP}}(\mathbf{f}_{1:k-1}|\mathbf{x}_{1:k-1}, \mathbf{y}_{1:k-1})d\mathbf{f}_{1:k-1}.$$

Splitting the integral into $\mathbf{f}_{1:k-2}$ and $\mathbf{f}_{k-1}$ we obtain:

$$p_{\text{RCGP}}(\mathbf{f}_k|\mathbf{x}_{1:k}, \mathbf{y}_{1:k}) \propto \exp(-\mathcal{L}(\mathbf{x}_k, \mathbf{y}_k, \mathbf{f}_k)) \int p(\mathbf{f}_k|\mathbf{f}_{k-1}, \mathbf{x}_{1:k}) \int p_{\text{RCGP}}(\mathbf{f}_{k-1}, \mathbf{f}_{1:k-2}|\mathbf{x}_{1:k-1}, \mathbf{y}_{1:k-1})d\mathbf{f}_{1:k-2}d\mathbf{f}_{k-1}$$

$$= \exp(-\mathcal{L}(\mathbf{x}_k, \mathbf{y}_k, \mathbf{f}_k)) \int p(\mathbf{f}_k|\mathbf{f}_{k-1}, \mathbf{x}_{1:k})p_{\text{RCGP}}(\mathbf{f}_{k-1}|\mathbf{x}_{1:k-1}, \mathbf{y}_{1:k-1})d\mathbf{f}_{k-1},$$

where in the last equality we integrate out $\mathbf{f}_{1:k-2}$ (which integrates to 1 since $p_{\text{RCGP}}$ is a density). Finally, using the inductive step assumption and the definition of the density $p_{\text{ST-RCGP}}$, we have:

$$p_{\text{RCGP}}(\mathbf{f}_k|\mathbf{x}_{1:k}, \mathbf{y}_{1:k}) \propto \exp(-\mathcal{L}(\mathbf{x}_k, \mathbf{y}_k, \mathbf{f}_k)) \int p(\mathbf{f}_k|\mathbf{f}_{k-1}, \mathbf{x}_{1:k})p_{\text{ST-RCGP}}(\mathbf{f}_{k-1}|\mathbf{x}_{1:k-1}, \mathbf{y}_{1:k-1})d\mathbf{f}_{k-1}$$

$$\propto p_{\text{ST-RCGP}}(\mathbf{f}_k|\mathbf{x}_{1:k}, \mathbf{y}_{1:k}),$$

Therefore, $p_{\text{RCGP}}(\mathbf{f}_k|\mathbf{x}_{1:k}, \mathbf{y}_{1:k}) \propto p_{\text{ST-RCGP}}(\mathbf{f}_k|\mathbf{x}_{1:k}, \mathbf{y}_{1:k})$. However, since both sides are densities, the proportionality implies equality; that is, $p_{\text{RCGP}}(\mathbf{f}_k|\mathbf{x}_{1:k}, \mathbf{y}_{1:k}) = p_{\text{ST-RCGP}}(\mathbf{f}_k|\mathbf{x}_{1:k}, \mathbf{y}_{1:k})$, which concludes the first step of the proof. Now, we want to show that the smoothing solutions are equal (claim 2.). We use a similar approach; however, the proof starts from the largest possible value of $k$ and then goes down in values.

**Step 2:** We want to show that $p_{\text{RCGP}}(\mathbf{f}_k|\mathbf{x}_{1:n_t}, \mathbf{y}_{1:n_t}) = p_{\mathcal{L}}(\mathbf{f}_k|\mathbf{x}_{1:n_t}, \mathbf{y}_{1:n_t})$.

Base Case:

Consider $k = n_t$. Then, the smoothing and filtering distributions are identically defined. Therefore, by **Step 1**, $p_{\text{RCGP}}(\mathbf{f}_{n_t}|\mathbf{x}_{1:n_t}, \mathbf{y}_{1:n_t}) = p_{\text{ST-RCGP}}(\mathbf{f}_{n_t}|\mathbf{x}_{1:n_t}, \mathbf{y}_{1:n_t})$.

Inductive Step:

Suppose that $p_{\text{ST-RCGP}}(\mathbf{f}_{k+1}|\mathbf{x}_{1:n_t}, \mathbf{y}_{1:n_t}) = p_{\text{RCGP}}(\mathbf{f}_{k+1}|\mathbf{x}_{1:n_t}, \mathbf{y}_{1:n_t})$ for $k \leq n_t - 1$. We want to show that $p_{\text{ST-RCGP}}(\mathbf{f}_k|\mathbf{x}_{1:n_t}, \mathbf{y}_{1:n_t}) = p_{\text{RCGP}}(\mathbf{f}_k|\mathbf{x}_{1:n_t}, \mathbf{y}_{1:n_t})$. Then, by Theorem 8.1 of Särkkä (2013), which requires that $\mathbf{f}_k$ be independent of $\mathbf{y}_{k+1:n_t}$ given $\mathbf{f}_{k+1}$ (satisfied by Markov assumption (iii)),

$$p_{\text{RCGP}}(\mathbf{f}_k|\mathbf{x}_{1:n_t}, \mathbf{y}_{1:n_t}) = p_{\text{RCGP}}(\mathbf{f}_k|\mathbf{x}_{1:k}, \mathbf{y}_{1:k}) \int \left[ \frac{p(\mathbf{f}_{k+1}|\mathbf{f}_k)p_{\text{RCGP}}(\mathbf{f}_{k+1}|\mathbf{x}_{1:n_t}, \mathbf{y}_{1:n_t})}{p_{\text{RCGP}}(\mathbf{f}_{k+1}|\mathbf{x}_{1:k+1}, \mathbf{y}_{1:k})} \right] d\mathbf{f}_{k+1}. \tag{22}$$

However, by **Step 1**, $p_{\text{RCGP}}(\mathbf{f}_k|\mathbf{x}_{1:k}, \mathbf{y}_{1:k}) = p_{\text{ST-RCGP}}(\mathbf{f}_k|\mathbf{x}_{1:k}, \mathbf{y}_{1:k})$, and the inductive step implies $p_{\text{RCGP}}(\mathbf{f}_{k+1}|\mathbf{x}_{1:n_t}, \mathbf{y}_{1:n_t}) = p_{\text{ST-RCGP}}(\mathbf{f}_{k+1}|\mathbf{x}_{1:n_t}, \mathbf{y}_{1:n_t})$. Therefore, there remains to show that $p_{\text{RCGP}}(\mathbf{f}_{k+1}|\mathbf{x}_{1:k+1}, \mathbf{y}_{1:k}) = p_{\text{ST-RCGP}}(\mathbf{f}_{k+1}|\mathbf{x}_{1:k+1}, \mathbf{y}_{1:k})$. But

$$p_{\text{RCGP}}(\mathbf{f}_{k+1}|\mathbf{x}_{1:k+1}, \mathbf{y}_{1:k}) = \int p(\mathbf{f}_{k+1}|\mathbf{f}_k)p_{\text{RCGP}}(\mathbf{f}_k|\mathbf{x}_{1:k}, \mathbf{y}_{1:k})d\mathbf{f}_k \tag{23a}$$

$$= \int p(\mathbf{f}_{k+1}|\mathbf{f}_k)p_{\text{ST-RCGP}}(\mathbf{f}_k|\mathbf{x}_{1:k}, \mathbf{y}_{1:k})d\mathbf{f}_k \tag{23b}$$

$$= p_{\text{ST-RCGP}}(\mathbf{f}_{k+1}|\mathbf{x}_{1:k+1}, \mathbf{y}_{1:k}), \tag{23c}$$

where in Equation (23a) we integrate and expand $p_{\text{RCGP}}(\mathbf{f}_k, \mathbf{f}_{k+1}|\mathbf{x}_{1:k+1}, \mathbf{y}_{1:k})$, in Equation (23b), we apply the result from Step 1, and in Equation (23c), we use the definition of $p_{\text{ST-RCGP}}$. We conclude that

Therefore, substituting the above quantities,

$$p_{\text{RCGP}}(\mathbf{f}_k|\mathbf{x}_{1:n_t}, \mathbf{y}_{1:n_t}) = p_{\text{ST-RCGP}}(\mathbf{f}_k|\mathbf{x}_{1:k}, \mathbf{y}_{1:k}) \int \left[ \frac{p(\mathbf{f}_{k+1}|\mathbf{f}_k) p_{\text{ST-RCGP}}(\mathbf{f}_{k+1}|\mathbf{x}_{1:n_t}, \mathbf{y}_{1:n_t})}{p_{\text{ST-RCGP}}(\mathbf{f}_{k+1}|\mathbf{x}_{1:k+1}, \mathbf{y}_{1:k})} \right] d\mathbf{f}_{k+1}$$
$$= p_{\text{ST-RCGP}}(\mathbf{f}_k|\mathbf{x}_{1:n_t}, \mathbf{y}_{1:n_t}).$$

This completes our inductive step and the proof. $\qquad \square$

### B.3. Proof of Proposition 3.3

To prove the robustness of ST-RCGP, we rely on the fact that ST-RCGP and RCGP share the same distribution for spatio-temporal data; therefore, we could use the following result from Altamirano et al. (2024) adapted to the spatio-temporal setting:

**Proposition B.2** (Altamirano et al. (2024)). *Suppose* $\mathbf{f} \sim \mathcal{GP}(m, k)$, $(\epsilon_1, ..., \epsilon_N)^\top \sim \mathcal{N}(0, \mathbf{I}_N \sigma^2)$, *and let* $C_k \in \mathbb{R}; k = 1, 2$ *be constants independent of* $y_{m,j}^c$. *Then, for RCGP regression with* $\sup_{\mathbf{x} \in \mathcal{X}, y \in \mathcal{Y}} w(\mathbf{x}, y) < \infty$ *has the PIF*

$$\text{PIF}(y_{m,j}^c, D) \le C_1 (w(\mathbf{x}_{m,j}, y_{m,j}^c)^2 y_{m,j}^c)^2 + C_2. \tag{24}$$

*Thus, if* $\forall \mathbf{x} \in \mathcal{X}$, $\sup_{y \in \mathcal{Y}} \left| y \cdot w(\mathbf{x}, y)^2 \right| < \infty$, *RCGP is robust since* $\sup_{y_{m,j}^c \in \mathcal{Y}} | \text{PIF}_{\text{RCGP}}(y_{m,j}^c, D)| < \infty$.

Hence, it suffices to verify that the proposed weight function satisfies the necessary conditions for robustness presented in Proposition B.2.

The weight function and the hyperparameter recommended are:

$$w_{\text{IMQ}}(\mathbf{x}, y) = \beta \left( 1 + \frac{(y - \gamma(\mathbf{x}))^2}{c(\mathbf{x})^2} \right)^{-\frac{1}{2}}, \tag{25}$$

with $\beta = \frac{\sigma^2}{2}$, $\gamma(\mathbf{x}) := \mathbb{E}_{p_{\mathcal{L}}}[y]$, and $c^2(\mathbf{x}) := \mathbb{E}_{p_{\mathcal{L}}}[(y - \gamma(\mathbf{x}))^2]$.

It is straightforward to verify that $w_{\text{IMQ}}(\mathbf{x}, y) \le \beta$ for all $\mathbf{x} \in \mathcal{X}$ and $y \in \mathcal{Y}$. Since $\beta = \frac{\sigma^2}{2} < +\infty$, it follows that $\sup_{\mathbf{x} \in \mathcal{X}, y \in \mathcal{Y}} w(\mathbf{x}, y) < +\infty$. Thus, with the recommended hyperparameters, $w$ satisfies the first condition.

Now, we need to check the second condition, which is that $\forall \mathbf{x} \in \mathcal{X}$, $\sup_{y \in \mathcal{Y}} |y| \cdot w(\mathbf{x}, y)^2 < +\infty$. To show this, let us consider an arbitrary $\mathbf{x} \in \mathcal{X}$ and two cases for $y$:

**Case 1:** $|y| \le |\gamma(\mathbf{x})| + |c(\mathbf{x})|$  Since $w_{\text{IMQ}}(\mathbf{x}, y) \le \beta$ for all $\mathbf{x} \in \mathcal{X}$ and $y \in \mathcal{Y}$, it follows:

$$|y| \cdot w(\mathbf{x}, y)^2 \le |y|\beta^2 \le \beta^2 (|\gamma(\mathbf{x})| + |c(\mathbf{x})|)$$

Which implies that, as long as $|\gamma(\mathbf{x})| < \infty$ and $|c(\mathbf{x})| < \infty$ we have that:

$$\sup_{\substack{y \in \mathcal{Y} \\ s.t.|y| \le |\gamma(\mathbf{x})| + |c(\mathbf{x})|\}}} |y| \cdot w(\mathbf{x}, y)^2 < \infty$$

**Case 2:** $|y| > |\gamma(\mathbf{x})| + |c(\mathbf{x})|$

$$|y| \cdot w(\mathbf{x}, y)^2 \le |y|\beta^2 \frac{1}{1 + \frac{(y - \gamma(\mathbf{x}))^2}{c(\mathbf{x})^2}} \le |y|\beta^2 \frac{1}{\frac{(y - \gamma(\mathbf{x}))^2}{c(\mathbf{x})^2}} = |y|\beta^2 \frac{c(\mathbf{x})^2}{(y - \gamma(\mathbf{x}))^2} = \beta^2 \frac{c(\mathbf{x})^2}{|y| \left( 1 - \frac{\gamma(\mathbf{x})}{y} \right)^2}$$

Now, we observe that this function is decreasing for $|y| > |\gamma(\mathbf{x})| + |c(\mathbf{x})|$, and particularly:

$$\lim_{|y| \to +\infty} \beta^2 \frac{c(\mathbf{x})^2}{|y| \left( 1 - \frac{\gamma(\mathbf{x})}{y} \right)^2} = 0,$$

since $|y| > |\gamma(\mathbf{x})|$. Therefore, attains its maximum at $|y| = |\gamma(\mathbf{x})| + |c(\mathbf{x})|$, which leads to the following bound:

$$|y| \cdot w(\mathbf{x}, y)^2 \leq \beta^2 \frac{c(\mathbf{x})^2}{|y| \left(1 - \frac{\gamma(\mathbf{x})}{y}\right)^2} \leq \beta^2 (|\gamma(\mathbf{x})| + |c(\mathbf{x})|).$$

Which implies that, as long as $|\gamma(\mathbf{x})| < \infty$ and $|c(\mathbf{x})| < \infty$ we have that:

$$\sup_{\substack{y \in \mathcal{Y} \\ s.t.|y| > |\gamma(\mathbf{x})| + |c(\mathbf{x})|\}}} |y| \cdot w(\mathbf{x}, y)^2 < \infty.$$

Finally, let us check that $|\gamma(\mathbf{x})| < \infty$ and $|c(\mathbf{x})| < \infty$. Since $\gamma(\mathbf{x}) = \mathbb{E}_{p_{\mathcal{L}}}[y]$ and $c^2(\mathbf{x}) := \mathbb{E}_{p_{\mathcal{L}}}[(y - \gamma(\mathbf{x}))^2]$ are the mean and the variance of $p_{\mathcal{L}}(y|\mathbf{x})$ respectively, and $p_{\mathcal{L}}(y|\mathbf{x})$ is a Gaussian, we know that the $|\gamma(\mathbf{x})| < \infty$ and $|c(\mathbf{x})| < \infty$.

Putting it all together, we have that:

$$\forall x \in \mathcal{X}, \sup_{y \in \mathcal{Y}} |y| \cdot w(\mathbf{x}, y)^2 < \infty.$$

## C. Additional Numerical Experiments

### C.1. Performance Metrics

Let $p_0(\cdot|\mathbf{x})$ be the density of the true data-generating process on the spatio-temporal grid $\mathbf{X} = (\mathbf{x}_1, \ldots, \mathbf{x}_k)^\top \in \mathcal{X}^{n_t}$. The first performance metric we use is the root mean squared error (RMSE):

$$\text{RMSE}(\mathbf{X}, \hat{y}) := \sqrt{\mathbb{E}_{Y \sim p_0(\cdot|\mathbf{X})}\left[(Y - \hat{y})^2\right]} \approx \sqrt{\frac{1}{n_t n_s} \sum_{k=1}^{n_t} \sum_{j=1}^{n_s} (y_{k,j} - \hat{y}_{k,j})^2},$$

where $N = n_t n_s$ is the number of data points, $y_i$ is the data, and $\hat{y}_i$ is the model's prediction. The second performance metric we use is the negative log predictive distribution (NLPD):

$$\text{NLPD}(\mathbf{X}, \hat{y}, \hat{\sigma}) := \mathbb{E}_{Y \sim p_0(\cdot|\mathbf{X})}\left[-\log p_\phi\left(Y|\hat{y}, \hat{\sigma}^2\right)\right] \approx -\frac{1}{N} \sum_{i=1}^{N} \log p_\phi(y_i \mid \hat{y}_i, \hat{\sigma}_i^2),$$

where $\hat{\sigma}_i^2$ is the model's variance on the prediction $\hat{y}_i$, and $p_\phi$ is the Gaussian density. Finally, for the methods with weights such as RCGP and ST-RCGP, we introduce the expected weight ratio (EWR):

$$\text{EWR}(\mathbf{X}) := \mathbb{E}_{Y \sim p_0(\cdot|\mathbf{X})}\left[\frac{w(\mathbf{x}, Y)}{w_{\text{STGP}}(\mathbf{x}, Y)}\right] \approx \frac{1}{n_t n_s} \sum_{k=1}^{n_t} \sum_{j=1}^{n_s} \left(1 + \frac{(y_{k,j} - \gamma(\mathbf{x}_{k,j}))^2}{c(\mathbf{x}_{k,j})^2}\right)^{-\frac{1}{2}},$$

where $w_{\text{STGP}} := \sigma/\sqrt{2}$ is the constant weight for the standard spatio-temporal GP. Note that by construction, $\text{EWR} \leq 1$. In particular, if $w_k = w_{\text{STGP}}$ for all $k = 1, ..., n_t$, then $\text{EWR} = 1$ and we recover the STGP posterior exactly. Therefore, since the STGP is not robust to outliers, EWRs that are near one are not necessarily optimal, since an EWR of 1 implies a solution—the vanilla STGP—which is not robust. This metric thus conveys a tradeoff between statistical efficiency—a model's ability to recover the STGP in well-specified settings—and robustness to outliers. In well-specified settings, we then want EWR to be larger and closer to one. When there are outliers, we wish the opposite so that we benefit from robustness.

### C.2. Implementation Details

All experiments are run on the CPU of a 2020 13-inch MacBook Pro with M1 chip and 8GB of memory.

**Definition of Matrices $\mathbf{A}_{k-1}$ and $\Sigma_{k-1}$** In the following, we provide a few examples of the SDE matrices needed to compute $\mathbf{A}_{k-1}$ and $\Sigma_{k-1}$, and explain how they are obtained in practice. We define $\mathbf{A}_{k-1} := \exp(\mathbf{F}\Delta t_k)$ and

$$\Sigma_{k-1} := \int_0^{\Delta t_k} e^{\mathbf{F}(\Delta t_k - \tau)} \mathbf{L} \mathbf{Q}_c \mathbf{L}^\top e^{\mathbf{F}(\Delta t_k - \tau)^\top} d\tau, \tag{26}$$

where $\mathbf{F} = \mathbf{I}_{n_s} \otimes \mathbf{F}_t \in \mathbb{R}^{n_s(\nu+1) \times n_s(\nu+1)}$, $\mathbf{L} = \mathbf{I}_{n_s} \otimes \mathbf{L}_t \in \mathbb{R}^{n_s(\nu+1) \times n_s}$, and $\mathbf{Q}_c = \mathbf{K}_s \otimes \mathbf{Q}_{c,t} \in \mathbb{R}^{n_s \times n_s(\nu+1)}$ for $(\mathbf{K}_s)_{ij} = \kappa_s(\mathbf{s}_i, \mathbf{s}_j)$. The notation $\otimes$ refers to the Kronecker product of matrices, and the term $\exp(\mathbf{F}\Delta t_k)$ corresponds to the matrix exponential. From Equation (26), we see that the matrices $\mathbf{A}_{k-1}$ and $\mathbf{\Sigma}_{k-1}$ for $k = 1, ..., n_t$ are derived from $\mathbf{F}_t$, $\mathbf{L}_t$ and $\mathbf{Q}_{c,t}$ in the SDE formulation.

**SDE Matrices for Common Kernels** We provide in Table 4 common kernels and their SDE matrices. That is, for each kernel we select, we specify the matrix form of $\mathbf{F}_t, \mathbf{L}_t$, along with $\mathbf{Q}_{c,t}$. We later explain how these can be used to compute $\mathbf{A}_{k-1}$ and $\mathbf{Q}_{k-1}$. The parameters in Table 4 are the lengthscale $\ell$, the amplitude $\sigma_\kappa$, and the period length $\omega_0$; the inputs of the kernels are $t, t' \in \mathbb{R}$ and $\tau := t - t'$; the functions used are the Gamma function $\Gamma$ and $K_\nu$ is the modified Bessel function of the second kind.

*Table 4.* Table of kernels and their corresponding SDE matrices

| Kernel | Formula | Parameters | $\mathbf{F}_t$ | $\mathbf{L}_t$ | $\mathbf{Q}_{c,t}$ |
|---|---|---|---|---|---|
| Wiener Process | $\kappa_{\mathrm{WP}}(t,t') := \sigma_\kappa^2 \min(t,t')$ | $\sigma_\kappa$ | $0$ | $1$ | $\sigma_\kappa^2$ |
| Exponential | $\kappa_{\exp}(\tau) := \sigma_\kappa^2 \exp(-\frac{\tau}{\ell})$ | $\sigma_\kappa, \ell$ | $-1/\ell$ | $1$ | $2\sigma_\kappa^2/\ell$ |
| Matérn $\nu = 1/2$ kernel | $\kappa_{\mathrm{Mat.}}(\tau) := \sigma_\kappa^2 \frac{2^{1-\nu}}{\Gamma(\nu)} \left(\frac{\sqrt{2\nu}\tau}{\ell}\right)^\nu K_\nu\left(\frac{\sqrt{2\nu}\tau}{\ell}\right)$ | $\sigma_\kappa, \ell; \lambda := \sqrt{3}/\ell$ | $\begin{pmatrix} 0 & 1 \\ -\lambda^2 & -2\lambda \end{pmatrix}$ | $\begin{pmatrix} 0 \\ 1 \end{pmatrix}$ | $4\lambda^3\sigma_\kappa^2$ |
| Matérn $\nu = 3/2$ kernel | (same as above) | $\sigma_\kappa, \ell; \lambda := \sqrt{5}/\ell$ | $\begin{pmatrix} 0 & 1 & 0 \\ 0 & 0 & 1 \\ -\lambda^3 & -3\lambda^2 & -3\lambda \end{pmatrix}$ | $\begin{pmatrix} 0 \\ 0 \\ 1 \end{pmatrix}$ | $16\sigma_\kappa^2\lambda^5/3$ |
| Periodic | $\kappa_{\mathrm{periodic}}(\tau) := \sigma_\kappa^2 \exp\left(-\frac{2\sin^2(\omega_0\tau/2)}{\ell^2}\right)$ | $\sigma_\kappa, \ell, \omega_0$ | $\sum_{j=1}^n \begin{pmatrix} 0 & -\omega_0 j \\ \omega_0 j & 0 \end{pmatrix}$ | $n\mathbf{I}_2$ | $\mathbf{0}$ |

The matrix $\mathbf{A}_{k-1}$ can be straightforwardly computed in most programming languages that offer linear algebra computations.

The covariance matrix $\mathbf{\Sigma}_{t,k-1}$ is rarely computed directly; instead, it is obtained by Matrix Fraction Decomposition (MFD) (see Chapter 6 of Särkkä & Solin (2019) for an overview of the method). Provided that the matrix $\mathbf{F}_t$ is Hurwitz, that is, all its eigenvalues have strictly negative real parts, the procedure to compute $\mathbf{\Sigma}_{t,k-1}$ can be further simplified (Särkkä & Solin, 2019) (i.e., no need for MFD) to

$$\mathbf{\Sigma}_{t,0} = \mathbf{\Sigma}_{t,\infty}$$
$$\mathbf{\Sigma}_{t,k-1} = \mathbf{\Sigma}_{t,\infty} - \mathbf{A}_{t,k-1}\mathbf{\Sigma}_{t,\infty}\mathbf{A}_{t,k-1}^\top,$$

where the initial covariance $\mathbf{\Sigma}_{t,0}$ can be found via the steady-state solution by solving for $\mathbf{\Sigma}_{t,\infty}$ in the following continuous Lyapunov equation (Särkkä & Solin, 2019; Hartikainen & Särkkä, 2010)

$$\mathbf{F}_t\mathbf{\Sigma}_{t,\infty} + \mathbf{\Sigma}_{t,\infty}\mathbf{F}_t^\top + \mathbf{L}_t\mathbf{Q}_{c,t}\mathbf{L}_t^\top = \mathbf{0}.$$

Note that we provide the full implementation for the ST-RCGP and the STGP in our code.

### C.3. Robust Hyperparameter Optimisation

**Issue With RCGP** Although the RCGP method is robust in inference, it still has well-known problems with hyperparameter optimisation when there are outliers. The method used is outlined in Altamirano et al. (2024), which corresponds to the leave-one-out cross-validation (LOO-CV). The optimisation objective is posed as follows:

$$\hat{\sigma}^2, \hat{\theta} := \arg\max_{\sigma^2, \theta} \left\{ \sum_{i=1}^n \log p^w(y_i|\mathbf{x}, \mathbf{y}_{-i}, \theta, \sigma^2) \right\},$$

where $\mathbf{y}_{-i} = (y_1, \ldots, y_{i-1}, y_{i+1}, \ldots, y_n)$. Moreover, the pseudo marginal likelihood is given by $p^w(y_i|\mathbf{x}, \mathbf{y}_{-i}, \theta, \sigma^2) = \mathcal{N}(\mu_i^R, \sigma_i^R + \sigma^2)$, where

$$\mu_i^R := z_i + \mathbf{m}_i - \left[\left(\mathbf{K} + \sigma^2\mathbf{J_w}\right)^{-1}(\mathbf{y} - \mathbf{m_w})\right]_i \left[\left(\mathbf{K} + \sigma^2\mathbf{J_w}\right)^{-1}\right]_{ii}^{-1}$$

$$\sigma_i^R := \left[\left(\mathbf{K} + \sigma^2\mathbf{J_w}\right)^{-1}\right]_{ii}^{-1} - \frac{\sigma^4}{2}w(x_i, y_i)^{-2},$$

where the notation follows that of Equation (8).

In Figure 7, we simulate data $y$ with outliers to highlight the issue. We generate 80 data points from a GP with Squared Exponential kernel of amplitude $\sigma_\kappa = 0.2$, lengthscale $\ell = 1$, and variance $\sigma^2 = 0.01$ in normally distributed data. We position four outliers around the $y = 2$ mark. To fit the data, we follow the suggested approach from Altamirano et al. (2024) and use a constant mean function equal to the arithmetic average of the data. In Figure 7, we show the results. Clearly, the RCGP fitting the data with outliers fails to produce adequate hyperparameters, whereas the fit on outlier-free data does well.

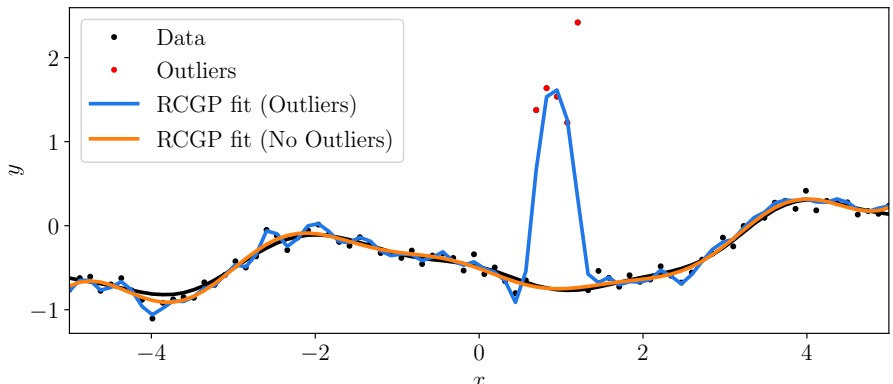

Figure 7. *RCGP Parameter Optimisation With and Without Outliers.* The fit with outliers yields a kernel lengthscale of 0.19, a kernel amplitude of 0.99, and variance of 0.026. The fit without outliers yields a kernel lengthscale of 1.46, a kernel amplitude of 36.9, and variance of 0.0066.

**Issue with STGPs Hyperparameter Optimisation**    the optimisation objective for STGPs is typically

$$\varphi(\boldsymbol{\theta}) := -\sum_{k=1}^{n_t} \log p(\mathbf{y}_k|\mathbf{y}_{1:k-1}, \boldsymbol{\theta}) = -\frac{1}{2}\sum_{k=1}^{n_t} \log|2\pi\mathbf{S}_k(\boldsymbol{\theta})| + \boldsymbol{\varepsilon}_k^\top(\boldsymbol{\theta})\mathbf{S}_k^{-1}(\boldsymbol{\theta})\boldsymbol{\varepsilon}_k(\boldsymbol{\theta}),$$

where $\mathbf{S}_k(\boldsymbol{\theta}) := \sigma^2\mathbf{I}_{n_s} + \mathbf{H}\mathbf{P}_{k|k-1}(\boldsymbol{\theta})\mathbf{H}^\top$ and $\boldsymbol{\varepsilon}_k := \mathbf{y}_k - \hat{\mathbf{f}}_k(\boldsymbol{\theta})$.

However, with outliers, this objective does not perform well since reasonable estimates of the latent function will have large $\boldsymbol{\varepsilon}_k$ and thus small $p(\mathbf{y}_k|\mathbf{y}_{1:k-1}, \boldsymbol{\theta})$—effectively fitting the outliers. This makes it so $\varphi$ may have a global minimum $\boldsymbol{\theta}^\star$ vastly different from the one in well-specified settings, in which case we could not recover the true latent function. We show in Figure 8 what happens when we fit STGP with a $\varphi$ objective. On the left side of the plot, we fit and make predictions with contaminated data. On the right side of the plot, we fit decontaminated data and make predictions on data with outliers. Both approaches yield undesired results, but most importantly, they are vastly different—a result that stems from introducing outliers in the data used for hyperparameter optimisation. Overall, Figure 8 shows that $\varphi$ is strongly affected by the presence of outliers and is not a robust objective for finding hyperparameters.

**Improving ST-RCGP's Hyperparameter Optimisation: Temporal Setting**    For temporal data, we now show that when there are outliers in the training data and we are using the ST-RCGP algorithm, choosing $\varphi_{\text{GB}}$ from Section 3 as an objective function for hyperparameter optimisation is more reliable than using the regular $\varphi$. It is given by:

$$\varphi_{\text{GB}}(\boldsymbol{\theta}) := \sum_{k=1}^{n_t} w_k \left( \log|2\pi\mathbf{S}_k(\boldsymbol{\theta})| + \boldsymbol{\varepsilon}_k^\top(\boldsymbol{\theta})\mathbf{S}_k^{-1}(\boldsymbol{\theta})\boldsymbol{\varepsilon}_k(\boldsymbol{\theta}) \right),$$

where definitions are as above with $n_s = 1$, and the weights $w_k$ are the ones from the ST-RCGP. In Figure 9, we fit the ST-RCGP on contaminated data with both objective functions to the best of our ability (that is, we adapt the learning rate and the number of optimisation steps as best as we can to get optimal results). We observe that using $\varphi_{\text{GB}}$ improves the optimisation process drastically compared to using $\varphi$, since way fewer steps were necessary, and doing so allows obtaining reasonable hyperparameter values for inference.

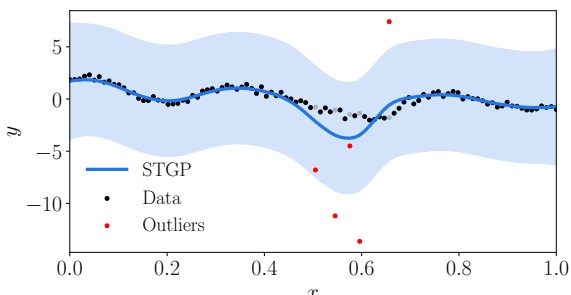

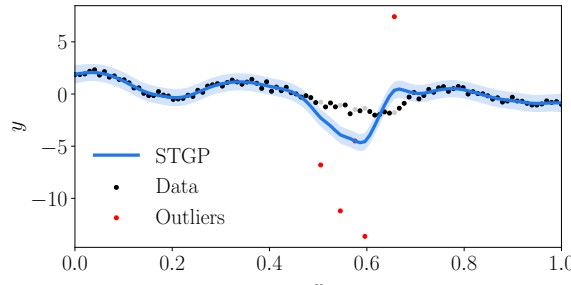

(a) *Fitting STGP on Outlier Data.* The hyperparameters obtained are: Lengthscale $\ell = 0.118$, amplitude $\sigma_\kappa = 3.438$, observation noise $\sigma^2 = 2.842$.

(b) *Fitting STGP on Outlier-Free Data.* The hyperparameters obtained are: Lengthscale $\ell = 0.236$, amplitude $\sigma_\kappa = 3.187$, observation noise $\sigma^2 = 0.055$.

Figure 8. *Impact of outliers on hyperparameter optimisation with $\varphi$.* The data is generated in the same way as for Figure 4 but with 100 data points instead and five outliers from a $\mathcal{N}(0, \sigma = 10)$. Both models use a Matérn 3/2 kernel.

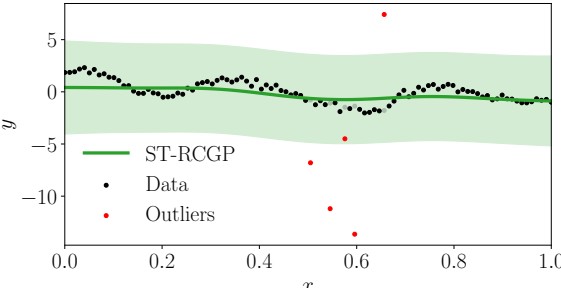

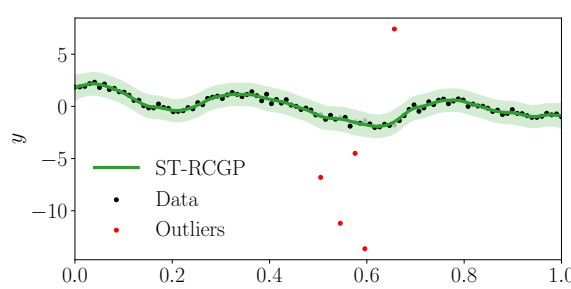

(a) *Fitting ST-RCGP on Outlier Data With $\varphi$.* The hyperparameters obtained are: Lengthscale $\ell = 0.683$, amplitude $\sigma_\kappa = 2.078$, observation noise $\sigma^2 = 1.909$. The optimisation process takes roughly 100 steps at a rate of 0.009 and is unstable (does not converge and produces infinite values if learning rate is increased).

(b) *Fitting ST-RCGP on Outlier Data With $\varphi_{GB}$.* The hyperparameters obtained are: Lengthscale $\ell = 0.112$, amplitude $\sigma_\kappa = 4.162$, observation noise $\sigma^2 = 0.094$. The optimisation process takes roughly 25 steps at a rate of 0.2 and is considerably more stable.

Figure 9. *Impact of objective function on ST-RCGP's hyperparameter optimisation.* The data is generated in the same way as for Figure 8. Both models use a Matérn 3/2 kernel and centering and shrinking function as specified in Section 3.

**Improving ST-RCGP's Hyperparameter Optimisation: Spatio-Temporal Setting** We turn our attention to spatio-temporal data and make the same point as in the previous paragraph, which is that choosing $\varphi_{GB}$ is more reliable than using the standard $\varphi$. We use the following objective:

$$\varphi_{GB}(\boldsymbol{\theta}) := \sum_{k=1}^{n_t} \tilde{w}_k \left( \log |2\pi \mathbf{S}_k(\boldsymbol{\theta})| + \boldsymbol{\varepsilon}_k^\top(\boldsymbol{\theta}) \mathbf{S}_k^{-1}(\boldsymbol{\theta}) \boldsymbol{\varepsilon}_k(\boldsymbol{\theta}) \right),$$

where now, the representation $\tilde{w}_k(\mathbf{w}_k)$ of the weights $\mathbf{w}_k$ at time step $t_k$ is given by:

$$\tilde{w}_k := \frac{Q_{k,n_s}(\delta)}{\sum_{i=1}^{n_t} Q_{i,n_s}(\delta)},$$

where $Q_{k,n_s}(\delta)$ is the $\delta$-quantile of the weights $\mathbf{w}_k$. In this experiment, we choose $\delta = 0.05$. The data we use is identical to that for Appendix C.4, except that the outliers are only introduced at steps $k = 2$ and 6, and the rest remains uncontaminated. As a performance metric, we use the cumulative mean absolute difference (CMAD) between our estimate of the latent function and the true latent function (not the observations), which is given by CMAD $= \sum_{k=1}^{n_t} \frac{1}{n_s} \sum_{j=1}^{n_s} |\mathbf{f}_{k,j} - \hat{\mathbf{f}}_{k,j}|$.

We fit the data with an ST-RCGP that has a Matérn 3/2 kernel, and a centering and shrinking function as specified in Section 3. We compare using $\varphi$ and $\varphi_{GB}$ and optimise for 30 training steps with an Adam optimiser with a learning rate of 0.1. We run the process five times with newly sampled data and outliers, keeping the latent function and the distributions from which the

data and outliers are sampled the same. We find that $\varphi$ yields CMAD values of $[1.9974, 1.6703, 1.9517, 1.6100, 1.7321]$, whereas $\varphi_{\text{GB}}$ produces CMAD values of $[0.4633, 0.5385, 0.5058, 0.4479, 0.5321]$. The $\varphi_{\text{GB}}$ objective yields substantially smaller CMAD values, and thus provides a more reliable hyperparameter objective function. We illustrate this in Figure 10, where we show three time steps of the first run (out of the five) for both objective functions.

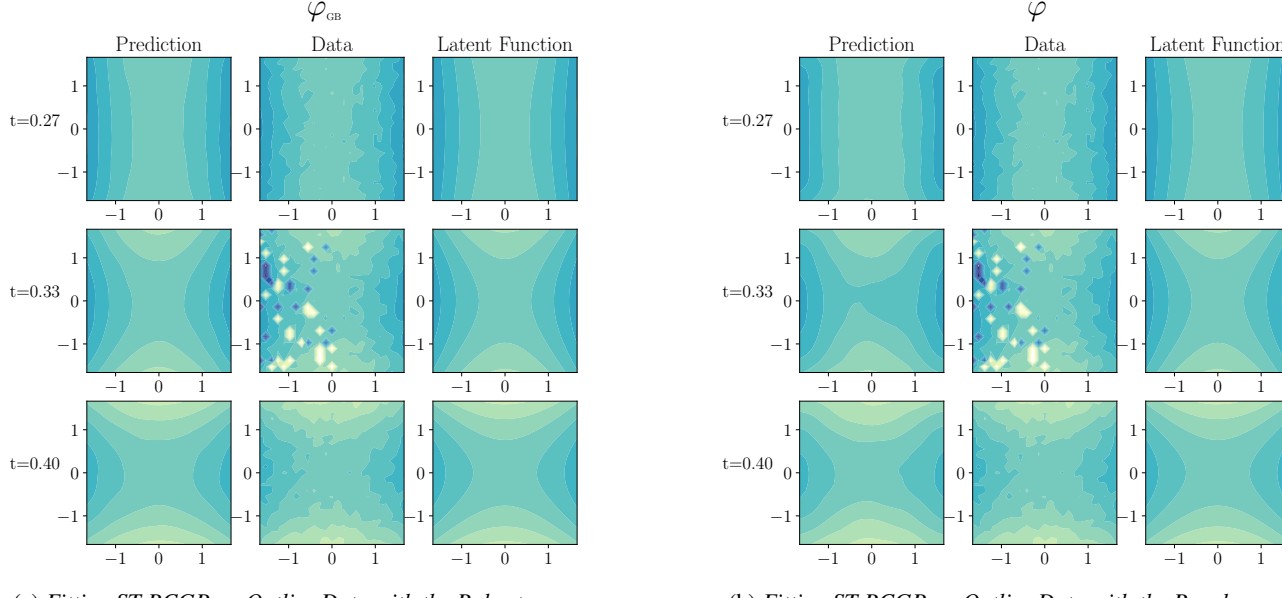

(a) *Fitting ST-RCGP on Outlier Data with the Robust $\varphi_{GB}$.*
(b) *Fitting ST-RCGP on Outlier Data with the Regular $\varphi$..*

Figure 10. *ST-RCGP Fits Using The Objective Functions $\varphi_{GB}$ and $\varphi$.*

## C.4. Synthetic Spatio-temporal Problem in Figure 1

A grid of size $n_s = 25 \times 25$ is produced and repeated through $n_t = 10$ time steps between $t_0 = 0.2$ and $t_f = 0.8$. We use a latent function $f(s_1, s_2, t) = \sin(2\pi t)s_1^2 + \cos(2\pi t)s_2^2$ and generate additive noise from a $\mathcal{N}(0, \sigma^2)$, where we set $\sigma = 0.2$. At any time step, if a data point is located in the region $s_1 < 0$, with $10\%$ probability, we contaminate the data point with an outlier sampled from $U([-8, -6] \cup [6, 8])$.

For both the STGP and ST-RCGP, we fit the data with the optimisation objective $\varphi$ and use de-contaminated data (original data without outliers) for the objective. We use the Adam optimiser with 20 training steps and 0.4 learning rate. The two algorithms use a Matérn 3/2 kernel. The ST-RCGP uses the adaptive centering and shrinking function from Section 3.

## C.5. RCGP Issues from Section 2

The data is generated by adding noise sampled from a $\mathcal{N}(0, \sigma = 0.5)$ to a latent function $f(x) = 3\sin(2\pi x)$ on a temporal grid $x \in [0, 1.4]$ with $n_t = 80$ points. We substitute at 8 locations outliers drawn from a $\mathcal{N}(3, \sigma = 0.2)$.

The RCGP optimisation process we use for kernel hyperparameters and observation noise is the one recommended in Altamirano et al. (2024). Note that we conduct optimisation on the original data without outliers. We have three configurations: First, with constant prior mean $m(x)$ equal to the data average and $c = Q_N(0.9)$; second, with $m(x) = \sin(2\pi x)$ and $c = Q_N(0.9)$; third, with $m(x) = \sin(2\pi x)$ and $c = 0.8$. All configurations use a Squared Exponential kernel and are separately optimised (that is, they do not necessarily share hyperparameters). Note that inference is conducted on data with outliers.

## C.6. ST-RCGP Posterior when Varying $c$ and $\beta$

We generate $N = n_t = 100$ data points from a GP prior with prior mean $m = 0$ and Matérn 3/2 kernel with $\ell = 0.2, \sigma_\kappa = 2.0$. We add noise from a $\mathcal{N}(0, \sigma^2)$ where $\sigma^2 = 0.25$. Outliers are generated by adding noise at 5 temporal locations drawn from a $\mathcal{N}(0, \sigma)$ with $\sigma = 20$.

The ST-RCGP uses a Matérn 3/2 kernel with lengthscale $\ell = 0.198$ and amplitude $\sigma_\kappa = 3.01$. We use an adaptive centering function for all values of $\beta$ and $c$. Then, we perform inference with varying values of $\beta$ and $c$ to conduct a sensitivity analysis of those hyperparameters. The results are shown in Figure 11.

We notice that increasing $\beta$ will lead to overfitting the data. Conversely, decreasing $\beta$ too much yields underconfident uncertainty estimates. The middle ground is $\beta = \sigma/\sqrt{2}$, which yields mean and uncertainty estimates that are appropriate, thus supporting our selection of hyperparameters.

For $c$, Figure 11 shows that a lower $c$ will be more robust to outliers, but may also overinflate the uncertainty estimate. On the other hand, a $c$ that is too large will produce an algorithm closer to the STGP, which is not robust to outliers. Visibly, the adaptive choice for $c$ performs the best, confirming our choice of hyperparameter.

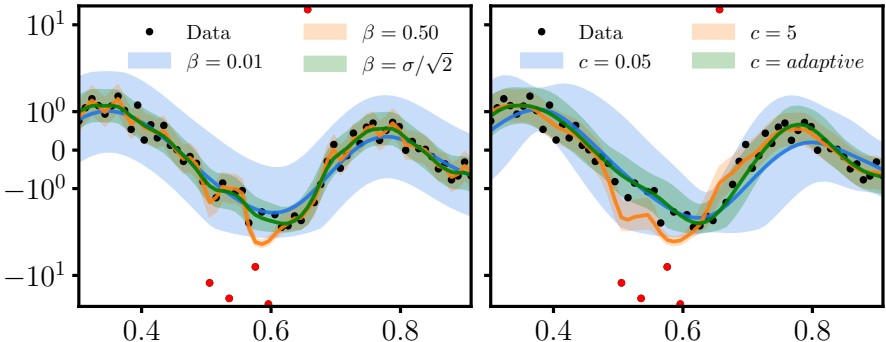

Figure 11. *Impact of $c$ and $\beta$ on the posterior.* We keep all parameters other than $\beta, c$ identical and choose $\mathbf{m}_k = \hat{\mathbf{f}}_k$. Outliers are highlighted in red. When $\beta$ increases (or decreases), the rate at which we learn from data increases (or decreases), and the confidence intervals narrow (or widen). When $c$ decreases, we increase robustness at the cost of larger posterior uncertainty. These tendencies support $\beta \propto \sigma$ and $c = \sqrt{\sigma^2 + \sigma_f^2}$ where $\sigma_f$ is the standard deviation of the predictive filtering posterior $p(y_k|y_{1:k-1})$.

### C.7. Well-Log Dataset

The well-log dataset, first introduced by Ruanaidh & Fitzgerald (1996), comprises 4,050 nuclear magnetic resonance measurements collected during the drilling of a well. Change points in the sequence indicate transitions between sediment layers encountered by the drill. In addition to these distinct transitions, the data also includes outliers and noise caused by shorter-term geological events, such as floods, earthquakes, or volcanic activity. In Figure 12, we contrast the results obtained from the ST-RCGP and the STGP. They perform equally well in well-specified regions, but the STGP lacks robustness to outliers.

### C.8. Sensitivity Analysis on IMQ Exponent

To understand how the shape of the weight function affects ST-RCGP's posterior estimates, we conduct a sensitivity analysis. We examine how varying the IMQ exponent—currently $\alpha := -1/2$— impacts results. We conduct this analysis because Proposition 3.3 shows robustness requires weights to decay faster than $1/\sqrt{|y|}$, i.e., $\alpha < -1/4$; but, overly fast decay can reduce statistical efficiency (overly robust)—highlighting a tradeoff worth exploring. The result of this analysis is shown in Figure 13.

### C.9. Sensitivity Analysis on Centering and Shrinking Functions

To capture the impact of the centering and shrinking functions on the posterior results, we conduct an analysis where we compare ST-RCGP and RCGP when all parameters are kept as in RCGP, apart from the centering function. We further explore the impact of also altering the shrinking function. To do so, we generate data with outliers the same way as in Appendix C.12. We then plot the posterior distributions of both algorithms in Figure 14 with a confidence interval (CI) of $3\sigma$.

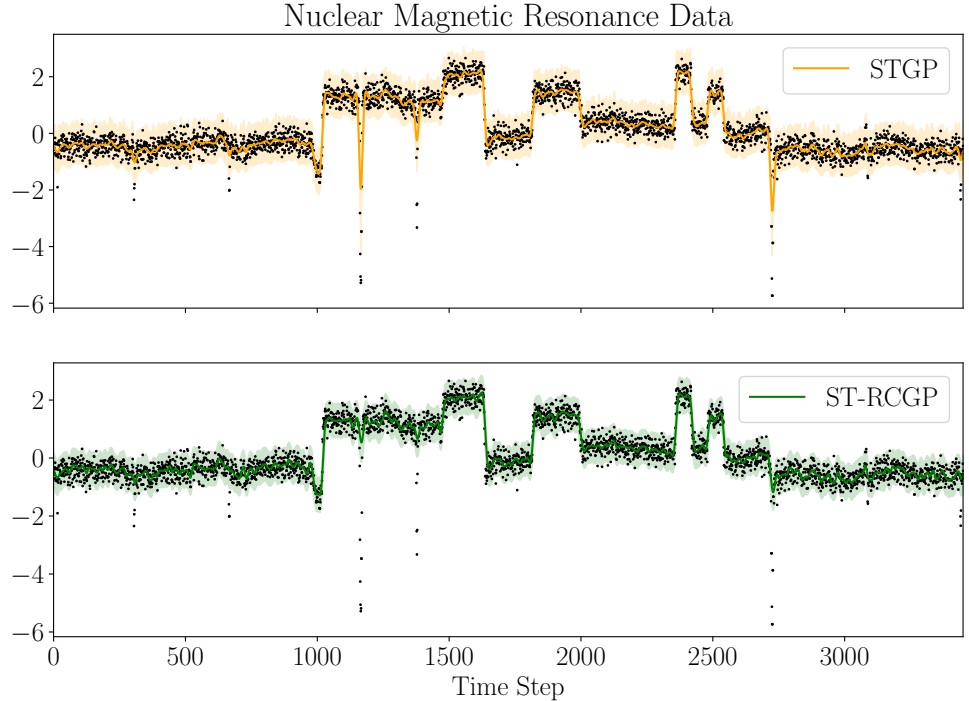

*Figure 12. Well-log data.* The top and bottom panels show the STGP and ST-RCGP fits to the data, respectively.

### C.10. Effect of Outlier Timing on ST-RCGP Posterior

We investigate the impact of changing the point in time at which outliers are introduced on the posterior of the ST-RCGP. The data is generated as in Appendix C.13. The results are shown in Figure 15.

### C.11. Choices of Centering Function in Special Cases

In Table 5, we highlight a few potential alternative choices other than ST-RCGP's filtering predictive $\hat{\mathbf{f}}_k$ for the centering function depending on whether the data is temporal or spatio-temporal. Also, we demonstrate which choices of centering functions recover the STGP and RCGP. The following points briefly explain the relevant features of each choice:

- Data: This choice, which yields constant weights and thus recovers the STGP, implies that our best estimate of the center of the data is the current observation. Interestingly, if the observation is an outlier, this implies we center the algorithm around the outlier. This provides an intuitive understanding for why the STGP fails to be robust.

- Prior Mean: Choosing for our centering function the prior mean $m(\mathbf{x}_k)$ recovers the RCGP (assuming a constant $c$ for the ST-RCGP). However, as previously explored in Section 4, this choice can yield poor results when the prior mean aligns with the outliers.

- Spatial Smoothing: This choice requires an additional hyperparameter (potentially many) that comes from $\bar{\mathbf{K}}_s$. The matrix $\bar{\mathbf{K}}_s$ dictates how much we want to smooth our data at time step $t_k$. This can be useful when there are unusual spatial structures in the data that $\bar{\mathbf{K}}_s$ can capture. Alternatively, when there are few time steps, the filtering predictive might not be the best estimate of the center of the data, in which case a good way to estimate the center that is more appropriate than a simple average is to use $\bar{\mathbf{K}}_s$.

- Temporal Smoothing: This centering function employs the same concept as the "Spatial Smoothing". It requires a lookback period $n_l$ and weights $\psi_i$ that determine how important the datum at step $i$ is to estimate the center of the data at step $k$.

- Filtering Predictive: This is the choice we make in this paper for ST-RCGP and explained in Section 3. It involves

Impact on ST-RCGP Posterior When Varying IMQ Exponent

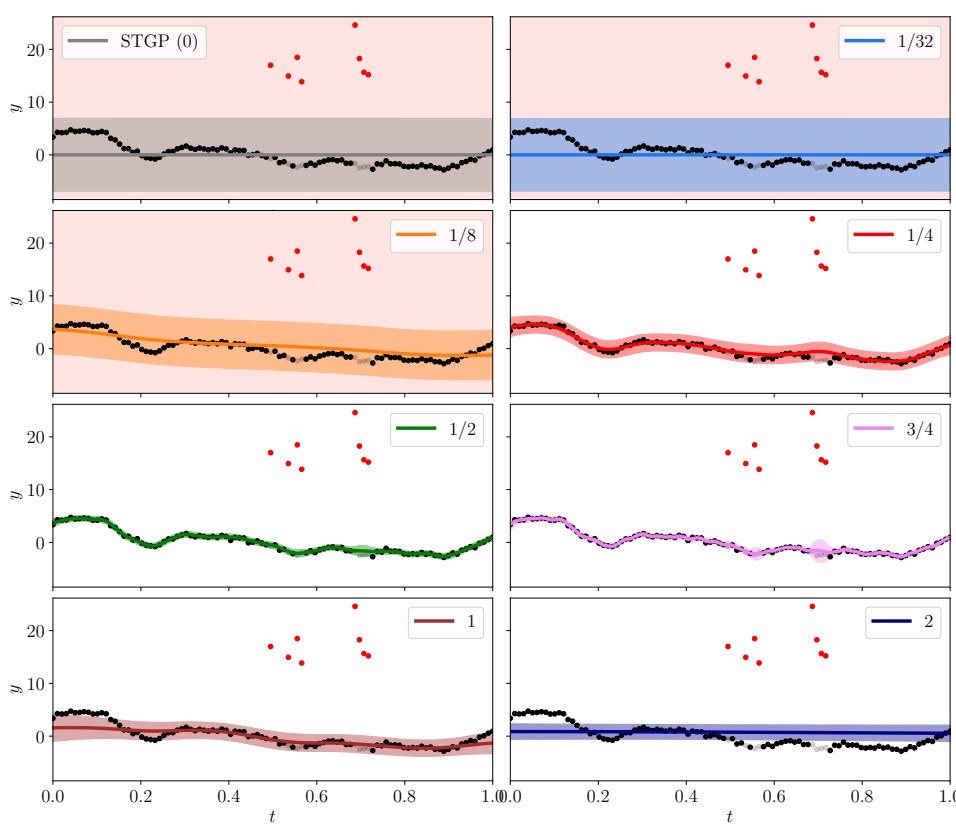

*Figure 13. Impact on ST-RCGP fit when varying the IMQ exponent.* The red-shaded plots indicate values of $\alpha$ that violate the robustness condition, i.e. $\alpha > -1/4$. The figures are denoted by $|\alpha|$, so that, for example, the top-right panel corresponds to $\alpha = -1/32$. Outliers are highlighted in red.

no additional hyperparameters, and is our model's best (and robust) prediction of $\mathbf{y}_k$ given past observations. These reasons are why we choose this option over the other for the ST-RCGP.

- Smoothed Predictive (S): Same concept as "Spatial Smoothing," but the smoothing is applied on predictions instead of on the data.

- Smoothed Predictive (T): Same concept as "Temporal Smoothing," but the smoothing is applied on predictions instead of on the data.

### C.12. Experiments Showing we Fix Vanilla RCGP from Section 4

**Data** We generate $n_t = 200$ data points at evenly spaced inputs in $x = [0, 1]$ from a GP with Matérn 3/2 kernel with lengthscale $\ell = 0.1$ and amplitude $\sigma_\kappa^2 = 2$, and mean function $m(x) = 2e^{-5x}$. The sample function $\mathbf{f} = (f_1, \ldots, f_{n_t})$ drawn is then centered: $\mathbf{f} \rightarrow \mathbf{f} - \bar{\mathbf{f}}$, where $\bar{\mathbf{f}} := \frac{1}{n_t} \sum_{i=1}^{n_t} f_i$. Noise is added and drawn from a $\mathcal{N}(0, \sigma^2)$ with $\sigma^2 = 0.25$. We contaminate the data with 10 outliers $|y_i^c|$ for $i = 1, ..., 10$ sampled from a $\mathcal{N}(5, \sigma^2 = 1)$. These outliers replace the data at specified locations. We keep both the original data and the contaminated data for further tasks.

**Fit** To fit the RCGP, we use the code from and follow Altamirano et al. (2024). We choose a constant weight function equal to the mean of the data, and $c = Q_N(0.95)$ since there are 5% outliers. The hyperparameters are optimized on the original data (without outliers) since training RCGP on contaminated data would result in overfitting the outliers and unreliable predictions. However, RCGP predictions are made on contaminated data.

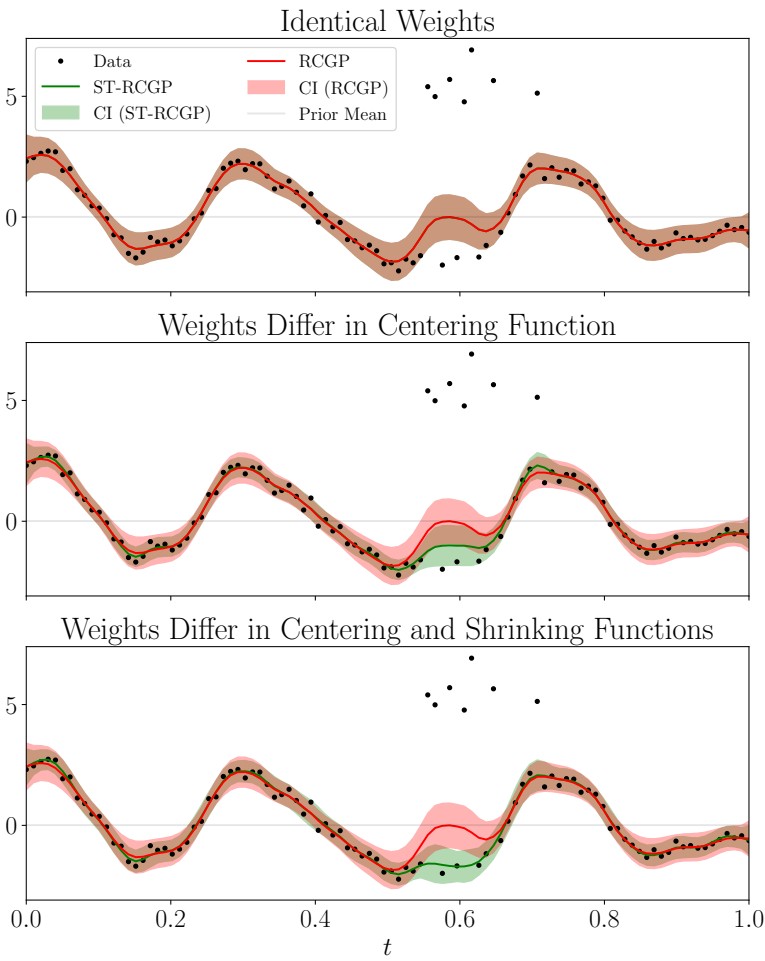

*Figure 14. ST-RCGP and RCGP comparison.* We contrast the two algorithms as we progressively change the specification of the weight function. The CI corresponds to the $3\sigma$ confidence interval. The prior is the constant function used in the weight function of the RCGP and the ST-RCGP in the first plot. In the second plot, the centering function $\gamma$ of the ST-RCGP is the predictive mean. In the third plot, the shrinking function $c$ is the filtering predictive's covariance, as specified in Section 3.

*Table 5. Choice of Centering Function $\boldsymbol{\gamma}_k$ For Weights $\mathbf{w}_k$.* $\psi_i$'s for $i = n_l, ..., k$ are normalised weights selected from the $n_l$-th past time step $t_{n_l}$ (lookback period). $\tilde{\mathbf{K}}_s \in \mathbb{R}^{n_s \times n_s}$ is a row-wise normalized kernel matrix.

| Algorithm | Centering $\boldsymbol{\gamma}(\mathbf{x})$ | Description |
|---|---|---|
| STGP | $\mathbf{y}_k$ | Data |
| RCGP | $m(\mathbf{x}_k)$ | Prior Mean |
| ST-RCGP | $\tilde{\mathbf{K}}_s\mathbf{y}_k$ | Spatial Smoothing |
| | $\sum_i \psi_i \mathbf{y}_i$ | Temporal Smoothing |
| | $\hat{\mathbf{f}}_k \equiv \mathbf{H}\mathbf{m}_{k|k-1}$ | Filtering Predictive |
| | $\tilde{\mathbf{K}}_s\hat{\mathbf{f}}_k$ | Smoothed Predictive (S) |
| | $\sum_i \psi_i \hat{\mathbf{f}}_i$ | Smoothed Predictive (T) |

To fit the ST-RCGP, we use the Adam optimiser and the robust scoring objective $\varphi_{\text{GB}}$ from Section 3. The hyperparameter selection is as in Section 3. Our learning rate is 0.3, and the number of optimisation steps is 70 (30 would be enough; we've

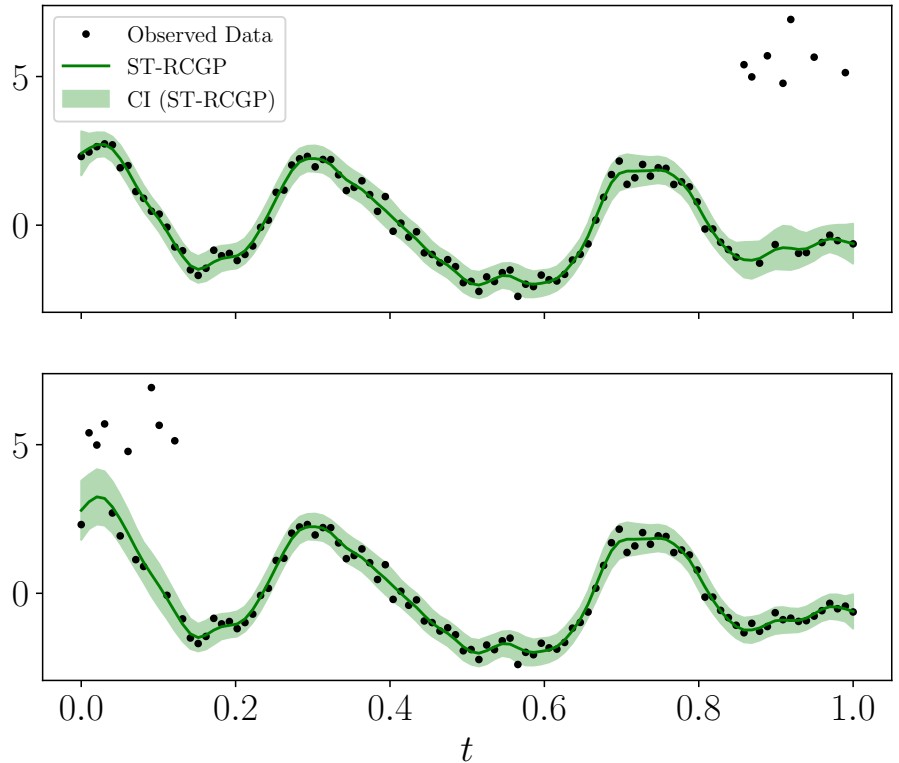

*Figure 15.* Impact of outlier timing on ST-RCGP Posterior.

used more to study convergence). In contrast to RCGP, the ST-RCGP is both training and predicting on contaminated data.

**Coverage** The coverage values are computed given a prediction $\mu$ and standard deviation $\sigma$. For each quantile, we find a corresponding $z$-score, and determine the proportion of data points falling within $\mu \pm z\sigma$.

### C.13. Experiments in Well-specified Settings from Section 4

We use the same dataset as in Appendix C.4, from which we can select the outlier-free data or the contaminated data. We use a Matérn 3/2 kernel for the GP prior. First, we perform hyperparameter optimisation on the dataset with outlier-free data. This involves 25 training steps using the Adam optimiser and a learning rate of 0.3. The criterion we use as our optimisation objective is the standard $\varphi$ (since there are no outliers). Second, we obtain the performance metrics for each model. This is done by generating new data as previously (from Appendix C.4). For each newly generated data, we compute statistical efficiency, RMSE and NLPD for each model. We take the average and standard deviation to report our metrics.

### C.14. Experiments with Financial Crashes from Section 4

**Twitter Flash Crash** We retrieve the "close" data from the DJIA index on April 23rd, 2013, and the previous day. This amounts to 810 data points. We build an evenly spaced temporal grid from 0 to 1 with 810 points. The observations are then standardised.

The GP fit is implemented in Python's *sklearn* package and uses a Matérn 3/2 kernel with amplitude $\sigma_\kappa = 0.72$, lengthscale $\ell = 0.0955$, observation noise $\sigma = 0.02$, and prior mean $m = 0$. The RCGP fit uses a Matérn 3/2 kernel with $\sigma_\kappa = 1$, lengthscale $\ell = 0.09$, a and a constant prior mean equal to the average of the data. Also, RCGP has $c = 0.25$, since it offers a more robust posterior than $c = Q_N$. The ST-RCGP fit uses a Matérn 5/2 kernel with amplitude $\sigma_\kappa = 1.$, lengthscale $\ell = 0.1$, $\sigma^2 = 0.02$ and has an adaptive shrinking and centering function, as specified in Section 3. The RP fit is exactly the one from Ament et al. (2024) since it is obtained using the same code.

The execution time is computed post-optimisation of each method, since we wish to capture execution time at inference. Also, to avoid caching and establish a fair comparison, each model has a second instance specifically for inference-making that hasn't observed data yet but has the optimised hyperparameters.

**Index Futures with Synthetic Crash** The data is obtained from https://www.kaggle.com/c/caltech-cs155-2020/data. It captures an Index Futures price over time, measured at 500ms intervals. We select $N = 46800$ data points, which amounts to a trading day. We built a temporal grid between 0 and 46800 and rescaled it by 0.5 ($500 \times 0.001$). The observations are then standardised. The crash induced aims to mimic the Twitter crash incident, but with slightly more outliers. Therefore, we drop 8 data points by $[0.9995, 0.9994, 0.9992, 0.996, 0.994, 0.998, 0.998, 0.9998]$ of their original value (not standardised) to create a V-shaped outlier region and add random noise to the drop sampled from a $\mathcal{N}(0, \sigma)$ for $\sigma = 0.0001$. Note that the amplitude of this drop is roughly similar to that of the Twitter flash crash experiment.

The STGP has a Matérn 3/2 kernel with $\ell = 6$, $\sigma_\kappa = 6$, and observation noise $\sigma = 0.14$. The ST-RCGP has a Matérn 3/2 kernel with $\ell = 6.5$, $\sigma_\kappa = 1$, and observation noise $\sigma = 0.3$. It also uses an adaptive centering and shrinking function as in Section 3. The *BayesNewton* methods all use a Student-T likelihood with df $= 6$ (degrees of freedom), except for the Laplace method, which has df $= 4$. They are also optimized following the code from Wilkinson et al. (2023).

The execution times are computed 5 times and evaluated on a new instance of the model's class to avoid caching issues. The one-step cost for each method is obtained by evaluating execution time for an increasing number of data points ($N = 5, 10, 100, 500, 1000, 2000, 2500, ..., 35000, 40000, 46800$), and then performing linear regression. The slope corresponds to the one-step cost. The RMSE and NLPD standard deviations are obtained by repeating inference on a newly generated crash event a total of 20 times.

In addition to Table 2, Figure 16 illustrates the fit of each method to the data.

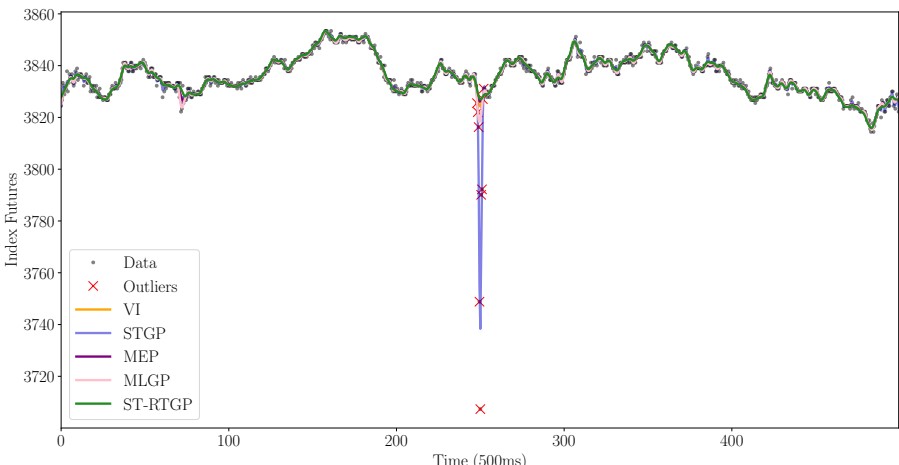

Figure 16. *Fitting the STGP, ST-RCGP and some methods from the BayesNewton package to the Index futures data with a synthetically induced crash.*

### C.15. Experiments with Temperature Forecasting from Section 4

The data is from the Climate Research Unit (CRU) and is available at https://crudata.uea.ac.uk/cru/data/hrg/. We select latitude and longitude ranges of $[45, 60]$ and $[-12, 8]$ respectively, which amounts to $n_s = 571$ spatial locations per time step. The data is monthly, starting in January 2022 and ending in December 2023, which is a total of $n_t = 24$ time steps. This leads to $N = n_t \times n_s = 11,991$ data points. We add 6 focussed outliers on a patch with latitudes and longitudes in $[51.25, 53.25], [-3, -1]$ respectively. The outliers are drawn from a $\mathcal{N}(120, \sigma^2)$ with $\sigma = 10$. Before fitting the data, we pre-process it by standardising.

To fit the data, we use the standard objective $\varphi$ (since the data has been cleaned by the CRU beforehand, see Harris et al. (2020)) with Adam optimiser, 60 optimisation steps and 0.05 learning rate. Both the STGP and the ST-RCGP use a Matérn 3/2 kernel for the temporal and spatial kernels. For STGP, this yields the following hyperparameters: A Temporal amplitude

of $\sigma_{\kappa_t} = 0.76$, temporal lengthscale $\ell_t = 2.51$, spatial amplitude $\sigma_{\kappa_s} = 0.76$, spatial lengthscale $\ell_s = 2.42$, and variance $\sigma^2 = 0.15$. For ST-RCGP, the hyperparameters are: A Temporal amplitude of $\sigma_{\kappa_t} = 1.12$, temporal lengthscale $\ell_t = 2.13$, spatial amplitude $\sigma_{\kappa_s} = 0.82$, spatial lengthscale $\ell_s = 2.34$, and variance $\sigma^2 = 0.069$. In Figure 17, we demonstrate the coverage of ST-RCGP and STGP on the temperature forecasting experiment for the month of October, which occurs before the introduction of outliers.

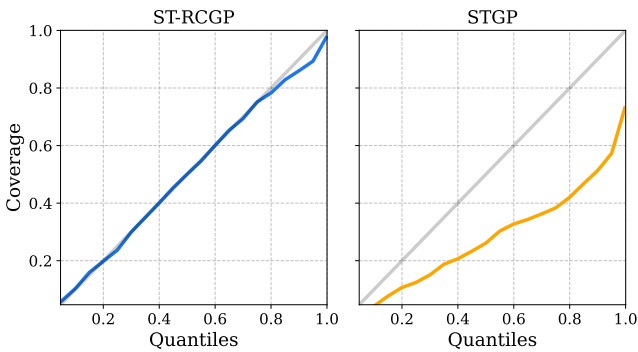

*Figure 17. STGP and ST-RCGP Coverage during the month of October.*

