# OpenReview forum: "Robust and Conjugate Spatio-Temporal Gaussian Processes"
_ICML.cc/2025/Conference — ICML 2025 poster_

### Official Review · Reviewer_Gtm6 · 2025-03-04

**Overall Recommendation:** 4

**Summary:**

The authors combine ideas from recent work on robust Gaussian processes with filtering ideas used in (spatio)temporal Gaussian process regression. They use temporal structure of the problem to set parameteres in the robust Gaussian process framework proposed by Altimirano et al 2024 (RCGP) in a sensible and automatic manner. They show that a computational speedup is possible using the filtering approach, allowing RCGPs to be scaled to spatio-temporal Gaussian processes with many time steps.

**Claims And Evidence:**

*Claim: The computational cost of the proposed method scale linearly in $N$ (made in paragraph state-space formulation)*

This claim is partially supported. Scaling is shown to be linear in the number of time steps (but not in the number of spatial locations). This claim is later stated correctly, and shown in proposition 3.1. The speed-up is also demonstrated in experiments. The others should avoid making the claim that the method scales linearly in the total number of observations, but the broader claim that the method improves scalability as compared to robust alternatives is well-supported.

*Claim: Selection of the prior mean is important in RCGP. The proposed method improves selection of the prior mean using temporal structure in the problem.*

Both parts of the claim, that RCGP is sensitive to selection of the prior mean, and that the proposed method, ST-RCGP, reduces this sensitivity, seem well-supported by the figure and argument in the text. Figure 3 supports the first part of this claim. I found the second part of the claim slightly less well-supported. It would be useful, though I do not think essential, if the authors ran an experiment in which the centering function was adaptive, but all other parameters were kept as in RCGP to show that this does in fact reduce sensitivity. I didn't see such an experiment in the supplement. If it is already there I would appreciate a pointer to it, and encourage the authors to reference it when discussing issue #1 in section 2.

*Claim: ST-RCGP improves upon (frequentist) coverage issues with RCGP*
This claim is supported by the coverage plot in figure 4. I think this claim has the least support. Consider including coverage plots for at least one other dataset, for example the temperature dataset considered. If this is not possible, could you explain why and consider including an additional simulation with spatial dimensions illustrating coverage properties of both methods?

**Essential References Not Discussed:**

I am not aware of essential references that were missed.

**Experimental Designs Or Analyses:**

I looked through the experimental design. Specific concerns were raised in a previous box on evaluation methods.

**Methods And Evaluation Criteria:**

The evaluation criteria and baseline make sense for the problem considered. Specific questions below:

*Figure 1 (description in appendix C.4), "For both the STGP and ST-RCGP, we fit the data with the optimisation objective φ and use de-contaminated data (original data without outliers) for the objective."* Why is decontaminated data used for selection of hyperparameters? Wouldn't it be more realistic to use the contaminated data?

*Timing in figure 5, details in appendix C.10 "The execution time is computed post-optimisation of each method, since we wish to capture execution time at inference.Also, to avoid caching and establish a fair comparison, each model has a second instance specifically for inference-making that hasn’t observed data yet but has the optimised hyperparameters.."*

I'm not convinced this is a meaningful timing comparison. It seems to me as though either optimization time should be included, as we are interested in the total cost of a procedure or caching should be used, because we only care about prediction at some new points after an algorithm has been trained. I would suggest including optimization time, or presenting both. Or could you clarify why the current approach might be meaningful for someone interested in using these methods in practice.

*Appendix C.10* are the parameter presented the parameter selected after optimization, or the initialized values of parameters?

**Other Comments Or Suggestions:**

I don't understand the caption of figure 2. What is mean't be "We emphasize..." and where can I see this in the figure?

**Other Strengths And Weaknesses:**

Presentation in the paper is generally very good. Problems are well-motivated.

**Questions For Authors:**

How does the point in time at which outliers occur affect the sensitivity of your method? In particular, it seems like at the first time step, your centering function doesn't improve on prior work since it relies on the filtering estimate. Does this mean the proposed method struggles more with outliers at early time steps than later time steps.

**Relation To Broader Scientific Literature:**

Generally, discussion of prior work was good. There should be a more clear description of the contribution relative to the prior work Duran-Martin et al, 2024. In particular, what is new? It seems the methodology in your paper is based on combining the Kalman filtering approach used in spatial GPs with the robust GP method proposed in Altamirano et al 2024. Initially I thought this was the main contribution. How does this compare to what was already done in Duran-Martin et al? Should I really understand the main contribution of the paper to be the selection of better weighting and centering functions? I think either is a reasonable contribution for a conference paper, given the high quality of presentation. But I'd like the scope of what is new, and how the paper builds on current work to be a bit clearer.

**Theoretical Claims:**

I looked over the proofs presented in appendix B. Although I did not go through them in sufficient detail to be confident of correctness, I did not see glaring issues, and the results seem very reasonalbe.

---

> ### Author Rebuttal · Authors · 2025-03-31
>
> We thank the reviewer for their careful consideration of our paper and helpful feedback. We address below the valuable suggestions made to further improve our work:
>
> **Scaling is shown to be linear in the number of time steps (but not in the number of spatial locations).**
>
> We agree with this concern. In the camera-ready version, we will replace the claim with “An alternative approach which has linear-in-time cost is to use a state-space representation.”
>
> **It would be useful (...) if the authors ran an experiment in which the centering function was adaptive, but all other parameters were kept as in RCGP to show that this does in fact reduce sensitivity.**
>
> We agree that isolating the effect of the adaptive centering function would strengthen our claim regarding reduced sensitivity. We will add an experiment comparing ST-RCGP and RCGP where all parameters are kept as in RCGP apart from the centering function. We further develop it by also altering the shrinking function. This analysis can be found here: https://anonymous.4open.science/r/ST-RCGP-21DD/tests/RCGP-vs-ST-RCGP-centering-function-sensitivity-analysis.ipynb. We welcome any further feedback.
>
> **Consider including coverage plots for at least one other dataset, for example the temperature dataset considered.**
>
> We agree that having more datasets for which we demonstrate the (frequentist) coverage of ST-RCGP would help further strengthen our claim. We will include in the paper a coverage analysis of the ST-RCGP for the temperature dataset. This analysis can be found at https://anonymous.4open.science/r/ST-RCGP-21DD/experiments/weather-forecasting/model-fitting.ipynb (at the end of the notebook).
>
> **Why is decontaminated data used for selection of hyperparameters? Wouldn't it be more realistic to use the contaminated data?**
>
> Fitting the RCGP on contaminated data leads to poor estimates due to overfitting outliers during hyperparameter optimisation. Using decontaminated data improves RCGP results. This choice thus strengthens our claim: even with this advantage, RCGP underperforms compared to ST-RCGP.
>
> **I'm not convinced this is a meaningful timing comparison.**
>
> We don’t include optimisation time because it can be easily skewed by tempering with the optimisation procedure, especially for STGPs, which rely on gradient-based optimisation without clear stopping criteria, number of optimisation steps, learning rate, and even optimisers. Either way, the two optimisation objectives (for STGP and ST-RCGP) are extremely similar, and their computational cost are nearly identical.
>
> Focusing on inference time is meaningful in online settings where one-step inference matters. For example, real-time applications like stock price estimation may preclude methods that iterate at each time step (e.g. variational STGP) due to time constraints, favoring closed-form solutions such as the Kalman filter, the robust filter from Durán-Martín et al. (2024), or the ST-RCGP.
>
> **Appendix C.10 are the parameter presented the parameter selected after optimization, or the initialized values of parameters?**
>
> The parameters presented are the ones post-optimization.
>
> **There should be a more clear description of the contribution relative to the prior work Duran-Martin et al, 2024. In particular, what is new?**
>
> The key differences are:
>
> 1) We deal with STGP problems such as hyperparameter optimisation and smoothing.
>
> 2) We carefully specify the weight function (see answer to vhF8 on IMQ) and its parameters to improve robustness (see Weight Function paragraph on line 237), whereas Duran-Martin et al. (2024) provide few justifications for the centering and shrinking functions and learning rate, which are parameters intrinsically connected to robustness and crucial to improving performance.
>
> 3) Duran-Martin et al (2024) use a weighted log-likelihood, while we use the weighted score-matching loss from Altamirano et al. (2024), leading to a different update step.
>
> We will include in the paper, at line 226, “However, these methods do not deal with STGP problems such as hyperparameter optimisation and smoothing and do not investigate downweighting optimally.”
>
> **I don't understand the caption of figure 2.” “Comment on appendix A about abuse of notation.**
>
> We thank the reviewer for bringing this to our attention. We will remove this sentence in the caption in the camera-ready version, and make clear that we are abusing notation.
>
> **How does the point in time at which outliers occur affect the sensitivity of your method?.**
>
> An empirical analysis we conducted, available at https://anonymous.4open.science/r/ST-RCGP-21DD/tests/outliers-early-vs-late.ipynb, demonstrates that the method is not affected by the impact of outlier position, which we expect to be because of smoothing.
>
> Finally, we want to thank the reviewer again for the careful review and consideration given to our paper. We hope this rebuttal addresses any remaining questions and concerns.

---

> > ### Comment · Reviewer_Gtm6 · 2025-04-03
> >
> > The authors addressed my questions regarding the experiments. The additional notebooks provide useful illustrations of different parts of the model. I am maintaining my score.

---

> > > ### Author Response · Authors · 2025-04-07
> > >
> > > We thank the reviewer for the thoughtful feedback and for mentioning the usefulness of the additional notebooks. We appreciate the decision to maintain the score.

---

### Official Review · Reviewer_vhF8 · 2025-03-10

**Overall Recommendation:** 4

**Summary:**

This paper introduces Spatio-Temporal Robust and Conjugate Gaussian Processes (ST-RCGPs), which are an extension of robust and conjugate Gaussian processes (RCGPs) [1]. ST-RCGPs leverage the state-space formulation of spatio-temporal GPs (STGPs) to achieve computational efficiency while maintaining the robustness properties of RCGPs. The paper also addresses three key limitations of RCGPs: sensitivity to prior mean misspecification, poor uncertainty quantification, and reliance on manual hyperparameter tuning. Robustness to outliers is established by showing that the posterior influence function is bounded. Experiments on synthetic, financial, and weather data compare ST-RCGP to STGP, RCGP, and other baselines in settings with outliers.

[1] Altamirano, M., Briol, F., Knoblauch, J. "Robust and Conjugate Gaussian Process Regression". ICML 2024.

---
## Update after rebuttal
The rebuttal has addressed my concerns and answered my questions. I have raised my score from 3 to 4.

**Claims And Evidence:**

The paper claims that
- ST-RCGP is an "outlier-robust spatio-temporal GP with a computational cost comparable to classical spatio-temporal GPs"
  - The robustness to outliers is demonstrated empirically (see e.g. Figures 4, 5, and 6, and Table 3) and theoretically (see Proposition 3.2). The improved computational efficiency is also demonstrated empirically (see e.g. Figure 5)
- ST-RCGP can "overcome the three main drawbacks of RCGPs: their unreliable performance when the prior mean is chosen poorly, their lack of reliable uncertainty quantification, and the need to carefully select a hyperparameter by hand.
  - This claim is addressed in Section 3 by adapting the weight function throughout filtering steps, whereas RCGPs require it to be fixed.

**Essential References Not Discussed:**

I am not aware of any essential references which were not discussed.

**Experimental Designs Or Analyses:**

The paper presents four different experiments:
1. "Fixing vanilla RCGP"
  - This experiment compares ST-RCGP to RCGP on 1D synthetic data to demonstrate that ST-RCGP addresses the three issues outlined  in Section 2 (sensitivity to prior mean, poor uncertainty quantification, selection of shrinking function necessary). The visualization (see Figure 4) looks convincing, although it is only a single example which could be selected on purpose.
2. "ST-RCGP in Well-Specified Settings"
  - This experiment compares ST-RCGPs to regular STGPs and RCGPs on synthetic data, demonstrating that ST-RCGP performs better than both other methods in the presence of outliers and matches performance otherwise. Since the data is synthetic, I have some concerns about how meaningful the results really are.
3. "Robustness During Financial Crashes"
  - This experiment is taken from [1] and [2]. It considers historical financial data and demonstrates that ST-RCGP is robust to a sudden crash which produces outliers in the time series. The experiment also demonstrates the superior computational efficiency of ST-RCGP compared to RCGP, which is obtained due to the state-space formulation. In a later part of this experiment, a larger version of the data with a synthetically induced crash is considered.
4. "Forecasting Temperature Across the UK"
  - This experiment focuses on temperature predictions and demonstrates that ST-RCGP performs well in the presence of outliers, whereas the forecasting of the regular STGP is strongly influenced by the outliers, which results in incorrect predictions. Again, the outliers are synthetically introduced.

While the presented results look quite strongly in favor of ST-RCGP, I am somewhat concerned about the amount of synthetic data used. Even the experiments involving real-world data have synthetically introduced outliers, which might not represent realistic scenarios.

[1] Altamirano, M., Briol, F., Knoblauch, J. "Robust and Conjugate Gaussian Process Regression". ICML 2024.

[2] Ament, S., Santorella, E., Eriksson, D., Letham, B., Balandat, M., Bakshy, E. "Robust Gaussian Processes via Relevance Pursuit". NeurIPS 2024.

**Methods And Evaluation Criteria:**

In terms of methods, the main idea is to combine the existing RCGP framework with state-space GPs for improved computational efficiency in spatio-temporal domains, which seems sensible to me. The fact that this allows for an adaptive weight function with additional benefits is great and also makes sense.

In terms of evaluation, the experiments seem to be somewhat contrived but reasonable to test the model. For example, the outliers introduced in the temperature forecasting experiment seem to be kind of unrealistic. The quantitative evaluation uses RMSE, NLPD, and compute time as metrics, which can be considered as standard in the literature.

**Other Comments Or Suggestions:**

N/A

**Other Strengths And Weaknesses:**

Strengths:
- Novel combination of robustness (RCGP) and efficiency (state-space formulation)
- Provides theoretical argument for robustness (Proposition 3.2)
- Thorough empirical evaluation on four different experiments

Weaknesses:
- The statement of Proposition 3.2 does not seem to be very strong
- Experiments make heavy use of synthetic data

**Questions For Authors:**

1. Do you have any additional experiments / empirical evidence which does not rely on synthetic data? If I understand correctly, only Figure 5 considers outliers from a real-world scenario.
2. Can you comment on any alternatives to the IMQ kernel for assigning the weights? This seems like a really important component for the method, yet the choice of IMQ seems somewhat arbitrary right to me.

**Relation To Broader Scientific Literature:**

The paper is primarily combines [1] with a state-space formulation of GPs. The resulting update equations are somewhat related to the Kalman filter. An important component of ST-RCGP (and also the original RCGP) is the inverse multi-quadratic kernel (IMQ) which is used to set the weights for each data point, such that outliers can (ideally) be assigned negligible contribution. In terms of the general goal of designing a robust GP, [2] is a recently published method with the same goal.


[1] Altamirano, M., Briol, F., Knoblauch, J. "Robust and Conjugate Gaussian Process Regression". ICML 2024.

[2] Ament, S., Santorella, E., Eriksson, D., Letham, B., Balandat, M., Bakshy, E. "Robust Gaussian Processes via Relevance Pursuit". NeurIPS 2024.

**Theoretical Claims:**

- Proposition 3.1 derives the state-space formulation of RCGPs. The resulting equations seem reasonable and the proof is provided in Appendix B.1, although I did not check the latter carefully.
- Proposition 3.2 shows that the posterior influence function is bounded (in its first argument) to demonstrate theoretical robustness. The assumptions seem to be mild, but the result is also not very strong (just because a quantity is not infinite it could still be really, or even arbitrarily, large...). The proof is provided in Appendix B.2., but I did not verify its correctness.

---

> ### Author Rebuttal · Authors · 2025-03-31
>
> We thank the reviewer for their thoughtful review and positive remarks on our empirical evaluation and the ST-RCGP’s novel combination of robustness and efficiency. We comment on the feedback below:
>
> **Since the data is synthetic, I have some concerns about how meaningful the results really are.**
>
> Robustness studies often inject outliers due to the lack of datasets with labelled outliers and the ability to control testing across varied outlier types (as was done in our work). However, we will add an experiment in the paper with the well-log dataset from [8, 11], which consists of 4,050 nuclear magnetic resonance measurements recorded while drilling a well. Outliers were identified scientifically and correspond to short-term events in geological history. The results can be found here: https://anonymous.4open.science/r/ST-RCGP-21DD/experiments/well-log-data/NMR-data-fit.pdf with the code in the same folder.
>
>
> **The statement of Proposition 3.2 does not seem to be very strong.**
>
> Whilst we empathise with this comment, the study of the influence function (IF) dates from the seminal work of Hampel in 1968 [1] and has been the gold standard in robust statistics ever since [2,3,4]. The typical approach to show robustness is to bound the influence function. While one could work out explicitly the bounds, there is no clear gain in doing so as the values of the KL divergence are not very interpretable. ​​To prove the robustness of an algorithm, bounding the IF is then sufficient and remains one of the strongest (and in many cases the only) available criteria within the robust Bayesian literature [see e.g. 5,6,7].
>
> **Can you comment on any alternatives to the IMQ kernel for assigning the weights?**
>
> We choose the IMQ for several reasons. To elaborate on those, in the final version, we change lines 237-243 with:
> “The statements of Propositions 3.1 and 3.2 impose some constraints on the choice of weight function. In particular, it should be strictly positive and differentiable over its domain to ensure quantities in Proposition 3.1 are well-defined, and it must have a bounded supremum to satisfy Proposition 3.2. Moreover, to avoid attributing weight to arbitrarily large $y\in \mathbb{R}$, we require $\lim_{|y| \rightarrow \infty}w(\mathbf{x}, y) = 0$ (decaying property). These requirements make the IMQ an appropriate choice of weight function, in addition to the fact that it has been well-studied and recommended in prior literature [6, 9, 10,11]. We further justify our choice on the basis that the IMQ hyper-parameters γ,$\beta$ and c are related to concepts of robustness—relations we exploit to specify the IMQ. In particular, to select γ, β and c, we follow four guiding principles; we want to: …”
>
> We acknowledge the need for a sensitivity analysis to understand how the shape of the weight function affects ST-RCGP's posterior estimates. In the camera-ready version, we will examine how varying the IMQ exponent—currently $\alpha:=-1/2$—impacts results. We conduct this analysis because Proposition 3.2 shows robustness requires weights to decay faster than $1/\sqrt{|y|},$, i.e., $\alpha<-1/4$; but, overly fast decay can reduce statistical efficiency (overly robust)—highlighting a tradeoff worth exploring. The result of this analysis can be found at https://anonymous.4open.science/r/ST-RCGP-21DD/tests/IMQ-sensitivity/IMQ-exponent-testing.pdf, with the code in the same folder.
>
>
> Thank you again for the thorough review. We hope this rebuttal addresses any remaining concerns and contributes to a stronger evaluation of our paper.
>
> [1] Hampel, Frank Rudolf. Contributions to the theory of robust estimation. 1968.
>
> [2] Huber, Peter J., and Elvezio M. Ronchetti. Robust statistics. 2011.
>
> [3] Maronna, Ricardo A., et al. Robust statistics: theory and methods (with R). 2019.
>
> [4] Hampel, Frank et al. Robust statistics: The approach based on influence functions. 1987.
>
> [5] Ghosh, A. and Basu, A. Robust Bayes estimation using the density power divergence. Annals of the Institute of Statistical Mathematics, 68(2), 2016.
>
> [6] Matsubara, T., Knoblauch, J., Briol, F.-X., and Oates, C. J. Robust generalised Bayesian inference for intractable likelihoods. Journal of the Royal Statistical Society: Series B (Statistical Methodology). 2022.
>
> [7] Duran-Martin, Gerardo, et al. "Outlier-robust Kalman filtering through generalised Bayes." 2024.
>
> [8] Altamirano, Matias, François-Xavier Briol, and Jeremias Knoblauch. "Robust and scalable Bayesian online changepoint detection." International Conference on Machine Learning. PMLR, 2023.
>
> [9] Chen, Wilson Ye, et al. "Stein point markov chain monte carlo." International Conference on Machine Learning. PMLR, 2019.
>
> [10] Riabiz, Marina, et al. "Optimal thinning of MCMC output." Journal of the Royal Statistical Society Series B: Statistical Methodology 84.4 (2022).
>
> [11] Ruanaidh, J. J. O. and Fitzgerald, W. J. Numerical Bayesian methods applied to signal processing. Springer Science & Business Media, 1996.

---

> > ### Comment · Reviewer_vhF8 · 2025-04-02
> >
> > I thank the authors for addressing my questions, and for providing additional explanations and empirical evidence. I have raised my score from 3 to 4.

---

> > > ### Author Response · Authors · 2025-04-07
> > >
> > > We thank the reviewer for the thoughtful feedback and for acknowledging the additional explanations and empirical evidence, as well as for raising the score from 3 to 4.

---

### Official Review · Reviewer_cvUj · 2025-03-12

**Overall Recommendation:** 4

**Summary:**

This paper introduces a methodology for spatio-temporal Gaussian Processes based on a state-space model and generalized Bayesian inference. Building on the robust and conjugate Gaussian processes (RCGPs) framework, it addresses and overcomes its key limitations, enhancing both robustness and computational efficiency.

**Claims And Evidence:**

The claims in this submission are well-supported by rigorous mathematical proofs and empirical experiments.

**Essential References Not Discussed:**

None.

**Experimental Designs Or Analyses:**

The experimental design looks good to me.

**Methods And Evaluation Criteria:**

The proposed methods are evaluated against both standard Gaussian Processes and the state-of-the-art RCGP framework, using multiple datasets and diverse evaluation metrics.

**Other Comments Or Suggestions:**

None.

**Other Strengths And Weaknesses:**

Strengths:
- Provides strong theoretical results.
- Evaluates the proposed ST-RCGP method across multiple experiments using various metrics.

Weaknesses:
- The baseline models used for comparison vary across experiments due to data characteristics. It would be beneficial for the authors to include an additional state-of-the-art model beyond RCGP to better assess the overall effectiveness of the proposed method.

**Questions For Authors:**

See above comments.

**Relation To Broader Scientific Literature:**

This paper extends the setting of previous work to spatio-temporal modeling by incorporating a state-space formulation, as discussed in prior literature. This formulation addresses three key limitations of the earlier approach.

**Theoretical Claims:**

Overall looks good to me.

---

> ### Author Rebuttal · Authors · 2025-03-31
>
> We thank the reviewer’s feedback and are glad our theoretical results and experimentation on our proposed method were appreciated. It has been mentioned that:
>
> **“The baseline models used for comparison vary across experiments due to data characteristics. It would be beneficial for the authors to include an additional state-of-the-art model beyond RCGP to better assess the overall effectiveness of the proposed method.”**
>
> We agree that it would be beneficial to illustrate the fit of additional methods apart from RCGP. For this reason, we will include in the paper the following plot: https://anonymous.4open.science/r/ST-RCGP-21DD/experiments/financial-applications/high-frequeny-data/HFT-data-induced-crash-fit.pdf, which was produced by code available here:
> https://anonymous.4open.science/r/ST-RCGP-21DD/experiments/financial-applications/high-frequeny-data/HFT-index-futures-speed-comp.ipynb. This plot compares ST-RCGP against some of the state-of-the-art methods offered by the “BayesNewton” package and is an extension of Table 2.
>
>
> Again, we are grateful for the consideration given to our paper and hope our answer satisfies the concern raised.

---

### Official Review · Reviewer_G2NE · 2025-03-14

**Overall Recommendation:** 4

**Summary:**

This paper expands the robust and conjugate GP framework to what it refers to as spatiotemporal GPs (sometimes referred to in other places as Markovian GPs, linear time GPs, or state space GPs). This is achieved by a generalized Bayes filtering solution, somewhat similar to other recent works on sequential generalized Bayes. Additionally, by considering the sequential nature of generalized Bayesian filtering, the authors are able to address some difficulties in "vanilla" robust and conjugate GPs that arise from finding appropriate weighting functions. The proposed method is tested on a number of synthetic and real-world datasets to show its effectiveness, both in spatiotemporal and 1-dimensional settings.

## Update after rebuttal

The authors have clearly answered my questions, softening their claims and pointing to related work where appropriate. In addition, I find the additional notebooks provided to other reviewers in the rebuttal process nice. I thus raised my score from a 3 to a 4 post-rebuttal.

**Claims And Evidence:**

The article makes a few main claims. The first is that the proposed generalized Bayes filtering solution is indeed a realization of the robust and conjugate GP framework. In particular, based on the chosen hyperparameters, it is claimed that the solution can be made robust (in the Huber sense), and that the vanilla GP can be recovered. A secondary set of claims state that several issues with vanilla RCGPs are ameliorated with some proposed changes to the weighting function. I find both of these sets of claims to be well-evidenced.

The only claim I find problematic is that the method "provides inferences that are comparable to state-of-the-art non-Gaussian STGPs in the presence of outliers, but at a fraction of the cost": Hamelijnck et al. show that their "parallel" formulation provide GPU implementations of MVGPs that are potentially orders of magnitude faster than the sequential formulation on CPU (c.f. Figure 4 in the Supplementary Material of [1]). To my knowledge, this formulation would not be possible with RC STGPs; the reviews indicate that all experiments are on CPU, in which case I find this claim a bit too general.

I also have some minor technical comments (see the points in "Other Comments or Suggestions" below)

**Essential References Not Discussed:**

Besides approximate or non-conjugate GPs, there is a small body of work using $3\sigma$-rejection-esque filters for Markovian GPs that are not cited [2, 3]; these can be seen as creating a more naive (and degenerate) weighting function. Importantly, both of these works are "adaptive" like the ST-RCGP, in the sense that the current posterior predictive moments are used sequentially to detect anomalies/outliers.

**Experimental Designs Or Analyses:**

I believe the experimental design and analysis are sound, besides the potential omission of the "parallel" variational STGPs as mentioned above.

**Methods And Evaluation Criteria:**

I believe the proposed methods and evaluation criteria make sense for the problem at hand. My only technical comment is regarding the EWR: it is stated that "since the STGP is not robust to outliers, the closer EWR is to one, the less robust our posterior is to outliers, and vice-versa." I don't think this statement is strictly accurate. For example, if one downweights in the "wrong" locations, robustness is actually decreased with the EWR. Perhaps a more correct statement is just that EWRs that are near one are not necessarily optimal, since an EWR of 1 necessarily implies a solution which is not robust. I understand the sentiment of this statement, however, and find it an intuitive and appropriate measure for experiments.

**Other Comments Or Suggestions:**

I have a few minor editorial comments:
1. (Line 082, Right Column) It is stated that "If standard GPs were used here, the cost would be $\mathcal{O}(n_t^3 n_s^3)$, which would be impractical," which I find unnecessarily strong. I suggest that this "may become impractical," instead.
2. (Line 224, Right Column) I think this should read "accomplishing the purpose of [principle] 3".
3. (Section 4) I understand that space constraints are an issue, but I think it would improve the exposition to motivate the EWR in a sentence or two, rather than just pointing the reader to a technical definition in the appendix.

And a small technical comment:
1. (Line 087, Right Column) It is written "assuming a stationary and separable kernel [...]" but stationarity is not sufficient for representation as an LTI SDE (e.g., the squared exponential kernel). Notably, however, any stationary kernel can be approximated arbitrarily well (c.f. [4]).

**Other Strengths And Weaknesses:**

### Strengths
- The paper is generally well-written and easy to follow.
### Weaknesses
- The paper might be seen as a straightforward combination of a few different ideas: STGPs and generalized Bayes filters. However, the benefits of adaptive weight centering and the thoroughness of experiments outweigh this weakness, in my opinion.

**Questions For Authors:**

1. At several points, the computational-aware RCGP is mentioned. [5] also takes a similar step in proposing computation-aware STGPs. Are there obvious technical barriers in also combining these works, i.e., having computation-aware ST-RCGPs?
2. How does the smoothing work? I see that there are smoothing solutions in the code, but it is not explained in the text. Does one obtain the "correct" smoothing solution by simply using the stored GB filtering statistics and applying RTS smoothing naively?

### References
[1] Hamelijnck, Oliver, et al. "Spatio-temporal variational Gaussian processes." Advances in Neural Information Processing Systems 34 (2021): 23621-23633.

[2] Bock, Christian, et al. "Online time series anomaly detection with state space Gaussian processes." arXiv preprint arXiv:2201.06763 (2022).

[3] Waxman, Daniel, and Petar M. Djurić. "A Gaussian Process-based Streaming Algorithm for Prediction of Time Series With Regimes and Outliers." 2024 27th International Conference on Information Fusion (FUSION). IEEE, 2024.

[4] Loper, Jackson, et al. "A general linear-time inference method for Gaussian Processes on one dimension." Journal of Machine Learning Research 22.234 (2021): 1-36.

[5] Pförtner, Marvin, et al. "Computation-aware Kalman filtering and smoothing." _arXiv preprint arXiv:2405.08971_ (2024).

**Relation To Broader Scientific Literature:**

Robust and scalable solutions to GPs are increasingly important. The submission does a good job of highlighting its relation to two main subsets of the GP literature: robust solutions -- particularly through generalized Bayes (such as RCGP), and scalable spatiotemporal solutions through state-space representations. Besides approximate or non-conjugate GPs, there is a small body of work using $3\sigma$-rejection-esque filters for Markovian GPs that are not cited [2, 3]; these can be seen as creating a more naive (and degenerate) weighting function. Importantly, both of these works are "adaptive" like the ST-RCGP, in the sense that the current posterior predictive moments are used sequentially to detect anomalies/outliers.

**Theoretical Claims:**

I have checked Proposition 3.1 carefully. My only concern is that the proof depends on a particular choice of $\mathcal{L}$, which is only briefly mentioned in the background section. I think it could avoid possible confusion to more clearly state that Proposition 3.1 assumes the modified Fisher divergence. I went through Proposition 3.2 more briefly, but did not spot any issues. It's not entirely clear to me how smoothing works (asked below).

---

> ### Author Rebuttal · Authors · 2025-03-31
>
> We appreciate the thoughtful feedback and the expertise on the matter. We are glad you found the paper clearly written and easy to follow. We comment on the feedback below:
>
> **1) “provides inferences that are comparable to state-of-the-art non-Gaussian STGPs in the presence of outliers, but at a fraction of the cost" (...) I find this claim a bit too general.**
>
> We acknowledge that our claim is too broad and will adjust it as follows:
> “The ST-RCGP provides inferences comparable to state-of-the-art non-Gaussian STGPs with a computational cost similar to classical STGPs.”
> Additionally, we believe further accelerating the ST-RCGP is an exciting direction for future work, so we will replace lines 422-426 (right side) with:
> “Our method builds on classical STGPs, which remain vulnerable to scaling in the spatial dimension and are not optimal if parallel computing is available. These issues were addressed by Hamelijnck et al. (2021) through variational approximations and parallel-scan algorithms (see also [1]), and could be adapted to ST-RCGPs, providing the same benefit.”
>
> **2) Regarding the EWR: it is stated that "since the STGP is not robust to outliers, the closer EWR is to one, the less robust our posterior is to outliers, and vice-versa." I don't think this statement is strictly accurate**
>
> We agree and will replace the highlighted sentence with: “Therefore, since the STGP is not robust to outliers, EWRs that are near one are not necessarily optimal, since an EWR of 1 implies a solution—the vanilla STGP—which is not robust."
>
> **3) I think it could avoid possible confusion to more clearly state that Proposition 3.1 assumes the modified Fisher divergence.**
>
> Thank you for raising this point. We will ensure to make this clearer in the camera-ready version by replacing the current text in Proposition 3.1 with: “Then, we obtain a (generalised) posterior (…) with the score-matching loss $\mathcal{L}$ given by X for Y” where X will be replaced by the loss as provided in lines 674-675, and Y the mathematical objects used in the loss, which are defined on line 677.
>
> **4) There is a small body of work using 3σ-rejection-esque filters for Markovian GPs that are not cited.**
>
> ​​Thank you for bringing this to our attention. We will include the following statement after the last sentence of the paragraph on line 066:
> “We also highlight a small body of work that uses outlier-rejection Kalman filters with STGPs to improve robustness in inference tasks such as prediction and anomaly detection. While these methods offer robustness at lower computational cost compared to non-conjugate STGPs, they are generally considered less expressive and not as strongly supported by theoretical foundations.”
>
> **5) minor editorial comments**
>
> We agree with these editorial comments and will make the necessary changes in the final version.
>
> **6) It is written "assuming a stationary and separable kernel (...)" but stationarity is not sufficient for representation as an LTI SDE (e.g., the squared exponential kernel).**
>
> Thank you for highlighting this. We will update the final version accordingly.
>
> **7) Are there obvious technical barriers in also combining these works, i.e., having computation-aware ST-RCGPs?**
>
> We believe there should be no fundamental technical barriers to combining these approaches because the filtering update in ST-RCGP closely mirrors that of the standard STGP. Given this structural similarity, the main arguments presented by Pförtner, Marvin, et al (2024) [2] should remain applicable.
>
> **8) How does the smoothing work?**
>
> The generalised Bayes approach used on the filtering distribution produces a Gaussian posterior, and thus, the resulting smoothing distribution remains Gaussian. Therefore, as for classical STGPs, we can apply RTS smoothing using the GB filtering posterior.
> To clarify ambiguities about smoothing, the above explanation will be added in the paragraph on lines 224-235 (Left side, Methodology section).
>
> Finally, we want to thank the reviewer again for the careful consideration given to our paper. We hope we have responded to any remaining questions or concerns in this rebuttal.
>
> **References**
>
> [1] Särkkä, Simo, and Ángel F. García-Fernández. "Temporal parallelisation of Bayesian smoothers." IEEE Transactions on Automatic Control 66.1 (2020): 299-306.
>
> [2] Pförtner, Marvin, et al. "Computation-aware Kalman filtering and smoothing." arXiv preprint arXiv:2405.08971 (2024).
>
> [3] Bock, Christian, et al. "Online time series anomaly detection with state space Gaussian processes." arXiv preprint arXiv:2201.06763 (2022).
>
> [4] Waxman, Daniel, and Petar M. Djurić. "A Gaussian Process-based Streaming Algorithm for Prediction of Time Series With Regimes and Outliers." 2024 27th International Conference on Information Fusion (FUSION). IEEE, 2024.

---

> > ### Comment · Reviewer_G2NE · 2025-04-02
> >
> > Thanks for the helpful reply. I have two remaining points before raising my score:
> >
> > > We acknowledge that our claim is too broad and will adjust it as follows: [...] Additionally, we believe further accelerating the ST-RCGP is an exciting direction for future work, so we will replace lines 422-426 (right side) with: “[...] and parallel-scan algorithms (see also [1]), and could be adapted to ST-RCGPs, providing the same benefit.”
> >
> > Thanks, I think the new claim is more appropriate. However, it is not obvious to me that parallel scans could be applied, at least not with the proposed weighting function -- parallel scans require associativity, and their application in Bayesian smoothing requires some quantities to be precomputed. Does the proposed (sequential) weighting not disrupt associativity/the ability to precompute all relevant quantities? I more or less agree that this would be possible using the original RCGP weighting function, but a core contribution of the current work is that this weighting function is problematic.
> >
> > > The generalised Bayes approach used on the filtering distribution produces a Gaussian posterior, and thus, the resulting smoothing distribution remains Gaussian. Therefore, as for classical STGPs, we can apply RTS smoothing using the GB filtering posterior. To clarify ambiguities about smoothing, the above explanation will be added in the paragraph on lines 224-235 (Left side, Methodology section).
> >
> > Thank you for the clarification. I understand that the filtering distribution is Gaussian and therefore RTS smoothing can be applied -- my question was moreso about the result. In particular, the paper claims that the smoothing solution would match the RCGP batch solution, but this isn't a priori obvious to me. Notably, I think this is different than the (still useful) claim that the filtering solution is correct and that the corresponding smoothing solution via RTS smoothing is robust/effective, which I do agree is well supported theoretically and experimentally.

---

> > > ### Author Response · Authors · 2025-04-07
> > >
> > > **1. Parallel-scan algorithms**
> > >
> > > Thank you very much for raising this point, which gave us the opportunity to dig deeper into this paper. It is not currently obvious to us that the sequential weights would disrupt associativity as used in this paper, although our limited expertise in this area means we have not been able to check this formally within the very small period of time available for this rebuttal. Since we are only mentioning parallel-scan algorithms as an interesting area of future work, we will instead temper our claim further in the conclusion as follows:
> > >
> > > “Our method builds on classical STGPs, which remain vulnerable to scaling in the spatial dimension and could be addressed similarly to Hamelijnck et al. (2021) through variational approximations. Furthermore, our method may be computationally sub-optimal if parallel computing is available, in which case parallel-scan algorithms (see also [1]) could be interesting to adapt to ST-RCGPs if possible”.
> > >
> > >
> > > **2. Smoothing Solution**
> > >
> > > Thank you, this is once again a really interesting point that we will clarify further in the final version of the paper. The batch implementation (as in standard RCGPs) and the sequential implementation (through Kalman filtering/smoothing as done in this paper) do lead to the same filtering and smoothing distributions assuming we use fixed (rather than adaptive) weights. We will add a proposition formally proving this in the camera-ready version of the paper. The gist of the argument is that if you take the smoothing distribution as expressed in Theorem 8.1 of [1] (In the “Bayesian smoothing equations” section of the 2013 version of the book), you can show inductively that each quantity in the expression is the same for the ST-RCGP and the RCGP, so long as a few conditions are met, such as the prior distribution and weight function being specified identically. Notably, this also requires the filtering distribution to match, which we will show in the added proposition. This being said, the reviewer is right that the adaptive weighting function proposed in this paper will result in two different distributions. We will clarify this point in the camera-ready version.
> > >
> > > To conclude, we would like to once again thank the reviewer for an engaging and in-depth discussion of our work that has further improved the manuscript.
> > >
> > > **References**
> > > [1] Simo Särkkä (2013). Bayesian Filtering and ¨ Smoothing. Cambridge University Press.

---

### Decision · Program_Chairs · 2025-05-01

**Decision:**

Accept (poster)

**Comment:**

This work builds upon the methodology in spatio-temporal Gaussian process (GP) models where the problem is cast into a state-estimation problem. The authors extend the so-called robust and conjugate GP framework to obtain an outlier-robust spatio-temporal GP model with a comparable computation cost to vanilla spatio-temporal GPs. This submission was reviewed by 4 reviewers in the field, and the paper ended up with an average overall recommendation of 4.0 (min: 4, max: 4). The average score increased from 3.5 -> 4.0 during the discussion. All reviewers argue for accepting the paper.

This paper has clear merits. All the reviewers appreciated the presentation and proposed methodology. On the other hand, the paper might be seen as a straightforward combination of a few different ideas. However, the benefits of adaptive weight centering and the thoroughness of experiments outweigh this weakness (quoting reviewer G2NE).